# Nanoformulated Zoledronic Acid Boosts the Vδ2 T Cell Immunotherapeutic Potential in Colorectal Cancer

**DOI:** 10.3390/cancers12010104

**Published:** 2019-12-31

**Authors:** Daniele Di Mascolo, Serena Varesano, Roberto Benelli, Hilaria Mollica, Annalisa Salis, Maria Raffaella Zocchi, Paolo Decuzzi, Alessandro Poggi

**Affiliations:** 1Laboratory of Nanotechnology for Precision Medicine, Fondazione Istituto Italiano di Tecnologia, 16163 Genoa, Italy; daniele.dimascolo@iit.it (D.D.M.); hilaria.mollica@iit.it (H.M.); Paolo.Decuzzi@iit.it (P.D.); 2Molecular Oncology and Angiogenesis Unit, IRCCS Policlinico San Martino, 16132 Genoa, Italy; sere_varesano@hotmail.it; 3Immunology Unit, Ospedale Policlinico San Martino, University of Genoa, 16132 Genoa, Italy; roberto.benelli@hsanmartino.it; 4CEBR, University of Genoa, 16132 Genoa, Italy; annalisa.salis@unige.it; 5Division of Immunology, Transplants and Infectious Diseases, IRCCS San Raffaele Scientific Institute, 20132 Milan, Italy; zocchi.maria@hsr.it

**Keywords:** Vδ2 T cells, zoledronic acid, polymeric nanoconstruct, anti-tumor immunity, colorectal carcinoma

## Abstract

Aminobisphosphonates, such as zoledronic acid (ZA), have shown potential in the treatment of different malignancies, including colorectal carcinoma (CRC). Yet, their clinical exploitation is limited by their high bone affinity and modest bioavailability. Here, ZA is encapsulated into the aqueous core of spherical polymeric nanoparticles (SPNs), whose size and architecture resemble that of biological vesicles. On Vδ2 T cells, derived from the peripheral blood of healthy donors and CRC patients, ZA-SPNs induce proliferation and trigger activation up to three orders of magnitude more efficiently than soluble ZA. These activated Vδ2 T cells kill CRC cells and tumor spheroids, and are able to migrate toward CRC cells in a microfluidic system. Notably, ZA-SPNs can also stimulate the proliferation of Vδ2 T cells from the tumor-infiltrating lymphocytes of CRC patients and boost their cytotoxic activity against patients’ autologous tumor organoids. These data represent a first step toward the use of nanoformulated ZA for immunotherapy in CRC patients.

## 1. Introduction

It is well established that the immune system is involved in controlling the growth and expansion of solid tumors, including colorectal carcinomas (CRCs) [1,2]. Tumor-infiltrating lymphocytes (TIL) comprise multiple cell subsets that can either kill tumor cells (effector anti-tumor lymphocytes) or modulate this response (regulatory T cells, Treg) [3,4]. A recent analysis of the overall survival of CRC patients revealed that the presence of intratumoral γδ T cells highly correlates with a significant favorable prognosis [3,4]. These cells are resident in the mucosal-associated lymphoid tissue of the gut epithelium and play a major role in controlling the integrity of the intestinal barrier when facing bacterial infections and pathogenic injuries [5,6,7,8,9]. Γδ T lymphocytes can be activated upon recognition of unprocessed non-peptide small molecules, including phosphoantigens (PA), derived from the mevalonate pathway in tumor cells [10,11,12]. Different drugs can be used to enhance γδ T cell activation, such as synthetic pyrophosphate-containing compounds [13,14]. Among these, aminobisphosphonates (N-BPs) are known to induce the activation and proliferation of the Vδ2 γδ T cell subset, thus increasing their blood concentration and anti-tumor activity [15,16,17,18]. For instance, the N-BP zoledronic acid (ZA) is a chemically stable analogue of inorganic pyrophosphate that inhibits the farnesyl pyrophosphate synthase (FPPS) of the mevalonate pathway and up-regulates isopentenyl pyrophosphate (IPP) accumulation, promoting the preferential growth of anti-tumor Vδ2 T cells in vitro and in vivo [19,20,21]. This immunostimulating property, together with the direct anti-tumor effects, have paved the way for clinical trials of different N-BPs [22,23,24,25,26]. We have recently reported that epithelial and mesenchymal cells from CRC specimens, upon exposure to ZA, stimulate the expansion of Vδ2 T cells with anti-tumor activity [21]. However, one of the main limitations in using free ZA for the treatment of CRC is its bone tropism. Indeed, N-BPs are used in the therapy of osteoporosis for their ability to reach the bone and activate the osteo-matrix deposition [16,17]. This property is a clear advantage when treating bone metastases of carcinomas or bone marrow tumors like multiple myeloma [16,17,18,19]. Nevertheless, the strong N-BPs’ concentration in the bone limits the distribution in other districts, such as a neoplastic mass developing in the colon. Furthermore, as with most small molecules, ZA has a rapid blood clearance that also limits its effective use in γδ T cell stimulation [27,28]. Therefore, novel delivery strategies are needed to capitalize on the multifaceted therapeutic properties of ZA. To improve the biodistribution and bioavailability of N-BPs, prodrugs have been proposed [29]. Although these have been shown to trigger the expansion of Vδ2 T cells more efficiently than the corresponding soluble form, these formulations do not warrant a significant reduction in bone tropism.

On the other hand, nanomedicine [30] could represent an ideal tool to enhance the circulation half-life and intra-tumor accumulation of ZA, while preserving its pharmacological features. Moreover, the effective use of nanoparticles for stimulating the immune system response against tumors has been recently demonstrated [31,32].

Nanoparticles (smaller than 200 nm) can exploit the presence of large fenestrations in the tumor vessels, the so-called enhanced permeation and retention effect (EPR), to increase drug accumulation in the neoplastic tissue compared to free drugs [33,34]. Moreover, as the molecules are loaded inside the nanoparticle, drug solubility and stability could be notably improved, preventing degradation by serum proteins. Finally, it is possible to control the release of the drug, hindering early leakage, and its natural accumulation in bone tissue [35,36,37].

In this work, ZA was reformulated into spherical polymeric nanoparticles (ZA-SPNs) with an aqueous core containing the active molecule, enclosed in a thin polymer matrix and a lipid monolayer. This architecture confers to ZA-SPNs an appearance similar to biological vesicles. These ZA-SPNs enhance the efficiency of ZA in the stimulation of CRC patients’ Vδ2 T cells that, in turn, kill autologous tumor organoids, supporting the proposal to use this drug formulation against CRC.

## 2. Results

### 2.1. Synthesis and Characterization of ZA-SPNs

In order to enhance tumor accumulation for future in vivo use, ZA was reformulated into nanoparticles. Briefly, spherical polymeric nanoparticles loaded with ZA (ZA-SPNs) were obtained via a double emulsion–evaporation process, modifying the single emulsion protocol that was previously described by Lee et al. [38]. This formulation comprised an aqueous core, entrapping the hydrophilic ZA molecule, confined within a polymer shell and an external lipid monolayer. A schematic representation of a ZA-SPN’s architecture is given in Figure 1A. Note that some of the lipids were conjugated with 2 kDa polyethylene glicole (PEG) chains, which limit unspecific absorption of blood proteins and extend the half-life of SPNs in circulation [35]. The ZA-SPNs were spherical in shape (Figure 1B), exhibited a hydrodynamic diameter of ≈170 nm and a polydispersity index (PDI) lower than 0.2, indicating a monodisperse particle population, and a negative ζ-potential of approximately −40 mV, similar to that of biological vesicles (Figure 1C). The nanoparticles were found to be stable at 37 °C for more than 2 weeks, preserving both the size and the monodispersity index (Figure 1D). This should be ascribed to the negative surface ζ-potential that prevented particle aggregation over time. Indeed, the ζ-potential was also stable over the same period of observation (Appendix A). Note that the geometry and shell core structure of ZA-SPNs make them similar to biological vesicles. The pharmacological properties of ZA-SPNs were analyzed in terms of ZA loading and release. Different input amounts of ZA were considered in formulating the nanoparticles, namely 50, 100, 300, and 800 μg. The best formulation was obtained for an initial input of 300 μg, which resulted in a final ZA concentration of ≈100 μM per batch (Figure 1E) and an encapsulation efficiency, defined as the weight percentage of the encapsulated ZA amount at the end of the process over the initial amount, of ≈10% (Appendix A). This was identified as the best configuration in that it maximized the ZA loading per batch, thus minimizing the mass of nanoparticles needed to deliver a given amount of ZA. The release of ZA from SPNs is given in Figure 1F, documenting a modest initial burst, corresponding to ≈25% ZA released after 4 h, followed by a sustained release, with more than 60% of ZA still entrapped in the SPNs after 2 days (50 h). Only after about 7 days (150 h) had all loaded ZA been released out of SPNs, as evaluated using high pressure liquid chromatography. This assured that the effects observed further in the experiments were not due to the early release of ZA, but to the one entrapped within the nanoparticles, and were working only after they were internalized.

### 2.2. ZA-SPNs Induce the Proliferation and Activation of Vδ2 T Lymphocytes

To test the ability of ZA-SPNs to trigger the expansion of Vδ2 T lymphocytes, peripheral blood mononuclear cells (PBMCs) of healthy donors were incubated with decreasing amounts of ZA-SPNs for 24 h. Then, cell cultures were supplemented with exogenous interleukin-2 (IL-2) and replaced with fresh IL-2 every three days. The amount of Vδ2 T cells was evaluated using immunofluorescence with an antibody specific for the Vδ2 chain of the T cell receptor (TCR), followed by florescence activated cell sorter (FACS) analysis, on days 0, 7, 14, and 21. The expansion rate of Vδ2 T lymphocytes were calculated by dividing the number of Vδ2 positive T cells at a given time point by their initial number. At the beginning, the amount of Vδ2 T cells present in the PBMCs was very low (2.5 ± 1.2% n = 10). As documented in Figure 2A (showing one representative experiment out of 10), both ZA-SPNs and soluble ZA were able to induce Vδ2 T cell proliferation, with a maximal expansion on day 21, although the effect of ZA-SPNs was more rapid than that of ZA. The second striking difference was the dose of ZA needed. Indeed, the maximum increase in the percentage of Vδ2 T cells (>90% of the whole cell culture) was reached in 21 days of culture with 0.05 μM of nanoformulated ZA (ZA-SPNs) against 1.0–5.0 μM of soluble ZA, depending on the PBMC donor (Figure 2B, mean ± SD of 10 donors). Specifically, the EC_50_ of ZA-SPNs was 1.355 × 10^−3^ µM, while that of ZA was 3.685 × 10^−1^ µM (Table 1). In other words, an almost 300 times higher soluble ZA dose was needed to obtain the same effect as ZA-SPNs. The expansion of Vδ2 T cells in response to ZA-SPNs was remarkable: the number of Vδ2 T cells rose from the initial value of 10^3^ to 23 × 10^6^ after 21 days, with a 23,000 fold increase (EC_50_ in Table 1); a similar effect was achieved with soluble ZA, only at two orders of magnitude higher concentrations (Figure 2B).

Due to the remarkable efficiency of ZA-SPNs, we further analyzed whether nanoformulated ZA, at the concentration of 0.05 µM (5 × 10^−2^ µM reported in Figure 1B), could trigger the proliferation of Vδ2 T cells, even without the addition of exogenous IL-2. Thus, PBMCs were labelled with carboxyfluorescein succinimidyl ester (CFSE), and on day 7 of the culture, proliferating Vδ2 T lymphocytes were identified using a double immunofluorescence assay, as cells with decreasing levels of CFSE content (green fluorescence intensity is inversely proportional to cell division rate) and reactivity with the anti-Vδ2 specific mAb. As shown in Figure 2C, a significant decrease of CFSE content in Vδ2 T cells was detected in ZA-SPNs cultures; in addition, an estimation of cell generations on day 7 showed that Vδ2 T cells incubated with ZA-SPNs had divided six to eight times, whereas cells treated with soluble ZA divided only one to two times (Figure 2D, ModFit LT analysis). At this time point, the percentage of Vδ2 T cells proliferating in ZA-SPN-stimulated cultures ranged from 20 to 35% in four out of five donors (Figure 2E); in contrast, no significant proliferation was detected with soluble ZA (Figure 2C–E) or empty SPNs (Figure 2E). This finding prompted the evaluation of whether ZA-SPNs could trigger the activation of Vδ2 T cells. It is well known that the neo-expression of activation antigens, such as cluster differentiation (CD) 25 and CD69, on T lymphocytes is a marker of cell triggering. Indeed, CD69 can function as a metabolic gatekeeper increasing glucose uptake [39], while CD25 is the α chain of the IL-2 receptor (IL2R) that increases the affinity for IL-2 of the βγ dimers of IL-2R [40]. Both these markers are involved in the growth of T lymphocytes [37,38]. CD25 expression on Vδ2 T cells, upon stimulation with ZA-SPNs, was detectable at 48 h, whereas no induction of CD25 was observed using soluble ZA or empty SPNs (Figure 2F). Neo-expression of the CD69 molecule on Vδ2 T cells was faster (detectable after 24 h) and stronger when PBMCs were incubated with ZA-SPNs compared to soluble ZA (Figure 2G,H). On the other hand, empty SPNs did not induce the expression of CD69 (not shown). Of note, the one donor whose Vδ2 T cells did not respond for proliferation (Figure 2E, light blue circle) also failed to express CD25 upon ZA-SPNs stimulation (Figure 2F, light blue circle). These data indicate that ZA carried by SPNs can deliver a strong activation signal to Vδ2 T cells, leading to the neo-expression of activation antigens and expansion in the absence of exogenous growth factors.

### 2.3. ZA-SPNs Mediated Vδ2 T Cell Expansion Is Associated with Isopentenyl-Pyrophosphate (IPP) Production

It is well established that monocytes (Mo) are the main peripheral blood cell population responsible for the N-BP-mediated triggering of Vδ2 T cells [6,20,21]. This is due to a monocyte’s ability to efficiently accumulate IPP when exposed to N-BPs, including ZA [6,20]. As such, we examined whether ZA-SPNs could stimulate Vδ2 T cell expansion from highly purified T lymphocytes co-cultured with Mo and analyzed after 21 days. Different T cell to monocyte (T:Mo) ratios were considered, ranging from 10:1 to 1000:1. ZA-SPNs triggered the optimal proliferation of Vδ2 T cells at the T:Mo ratio of 10:1–100:1 (10^5^ T vs. 10^3^–10^4^ Mo) (Figure 3A). Notably, the EC_50_ for ZA-SPNs was 5.644 × 10^−4^ µM and 3.527 × 10^−1^ µM for ZA using 10^4^ Mo, which was more than 600-fold higher compared to ZA-SPNs (Figure 3A vs. Figure 3B). With 10^3^ Mo, EC_50_ became 7.096 × 10^−4^ µM and 5.449 × 10^−1^ µM for ZA-SPNs and ZA, respectively; in this case, the difference was larger than 700 times in favor of ZA-SPNs (Table 1 and Figure 3A vs. Figure 3B). This demonstrated that ZA-SPNs efficiently stimulated Vδ2 T cell proliferation in the presence of very few Mo, with a strong gain in EC_50_. Furthermore, it was assessed whether ZA-SPNs could stimulate the production of IPP, which is considered the product responsible for ZA-mediated effects [10,11,12,21]. Toward this aim, highly purified peripheral Mo were incubated with either ZA-SPNs (0.5 µM for 10^6^ cells needed for IPP evaluation) or soluble ZA (0.5 µM for comparison with ZA-SPNs at the same concentration) for 24 h. The IPP concentration was determined using high performance liquid chromatography/time of flight-mass spectrometry (HPLC/TOF-MS). ZA-SPNs induced a twofold higher IPP production than soluble ZA, with a mean value of 3 pM for ZA-SPNs against 1.5 pM for ZA in three out of four Mo donors (Figure 3C). Then, we tested whether IPP could also be produced by CRC cell lines in response to ZA-SPNs. Indeed, this event is needed to trigger Vδ2-T-cell-mediated recognition and killing [10,11,12]. For this experiment, LS180 and SW620 were chosen, since we reported that the two cell lines produce high or low amounts, respectively, of IPP when stimulated with soluble ZA [21]. LS180 produced about 30% more IPP in response to 0.5 µM ZA-SPNs compared to soluble ZA (14.12 pM vs. 9.92 pM), while SW620 IPP production to 0.5 µM ZA-SPNs was superimposable (≈3 pM) with that obtained with soluble ZA at the same dose (Figure 3D,E). A small but detectable IPP production was also observed in LS180 using ZA-SPNs at a 0.05 µM concentration (5 pM). Empty SPNs did not evoke the production of IPP (<1 pM, Figure 3D,E).

### 2.4. CRC Cell Death Induced by Vδ2 T Lymphocytes Stimulated with ZA-SPNs

At first, the interaction of Vδ2 T lymphocytes with tumor cells challenged with ZA-SPNs was qualitatively analyzed through confocal microscopy and SEM imaging. SW620 CRC cells were incubated with Cy5-labeled ZA-SPNs to prove that nanoparticles could be internalized by tumor cells. Figure 4A shows that Cy5-ZA-SPNs (orange, arrows) accumulated in CRC cells, evidenced with the anti-epithelial cell adhesion molecule (EPCAM) mAb (red), in agreement with the notion that nanoparticles of this size are rapidly (by 4 h) engulfed by any cell, even if not professional phagocytes [41,42,43]. Then, Vδ2 T cells were added to CRCs engulfed with ZA-SPNs for a short incubation time (further 4 h), and their ability to interact with CRC cells was evaluated. Figure 4B shows an intimate contact between Vδ2 T cells, identified using the specific anti-Vδ2 mAb (green) and CRC (red); arrows indicate the engulfed Cy5-ZA-SPNs. Moreover, Vδ2 T cells also appeared clustered on CRCs, as it can be better appreciated in the representative SEM image (Figure 4C) where several Vδ2 T cells, false-colored in pink, surround and make contact with a much larger cancer cell.

Having shown that ZA-SPNs can be metabolized by CRC cells, and that Vδ2 T lymphocytes bind to ZA-SPN-treated CRC, we further analyzed whether these events resulted in an anti-tumor effect. Toward this aim, Vδ2 T lymphocytes were added to CRC cell lines exposed to ZA-SPNs at an effector:target (E:T) ratio of 3:1 to for allow the detection of the effect elicited by the nanoformulated ZA. Indeed, at this E:T ratio, the cytotoxic activity of Vδ2 T cells to the three CRC cell lines tested (SW620, HCT-15, and HT-29), evaluated as a reduction of living cells by the crystal violet assay, was almost negligible (less than 10%; see the cell viability ranging from 90 to 100% with the lowest ZA-SPNs concentrations in Figure 4D). The percentage of living CRC cells was strongly reduced in the presence of Vδ2 T cells when serial dilutions of ZA-SPNs were added to the lymphocyte–CRC cell co-cultures (Figure 4D). The ZA-SPNs’ EC_50_ of cytotoxic activity was again lower (by 10 to 50 times) than that found for soluble ZA (Table 2). Both ZA-SPNs and ZA, in the absence of Vδ2 T cells, reduced the percentage of living adherent SW620 and HT-29 cells to some extent; this effect was more evident on HCT-15 (Appendix A), but was always lower than that detected in the presence of Vδ2 T lymphocytes (EC_50_ 0.5 µM in Appendix A vs. 0.1 µM depicted in Figure 4A and reported in Table 2).

Since tumor cells grow in vivo as tridimensional (3D) structures, tumor spheroids have been used to mimic the initial stage of neoplastic growth in vitro [44,45]. Thus, we assessed whether Vδ2-T-cell-mediated killing of such structures could be stimulated by ZA-SPNs. We analyzed the tumor spheroids obtained after culturing the representative three CRC cells lines SW620, HCT-15, and HT-29 in ultra-low attachment plates, as described in Varesano et al. [45]. In these experimental conditions, spheroids from 50 to 250 µm in diameter can be obtained [45]. CRC tumor spheroids were able to engulf ZA-SPNs within 24 h of exposure (Appendix A) and were killed by Vδ2 T cells; the ZA-SPNs EC_50_ was superimposable or lower than that of ZA for HCT-15 and HT-29 cell lines, while for SW620, the EC_50_ of ZA-SPNs was higher (Figure 4E and Table 2). A possible explanation for the discordant effect of ZA-SPNs and soluble ZA on SW620 as adherent cells or spheroids could be related to the different nanoparticle processing and/or mevalonate metabolism that follows the ZA-SPNs’ entry, despite the detectable engulfment. The direct effect of ZA-SPNs on CRC cells growing adherent in conventional experimental conditions or cultured as spheroids was also evaluated using a crystal violet assay. Both ZA-SPNs and soluble ZA reduced the number of living cells by 30% when used on SW620 and HT-29 CRC cell monolayers, while the HCT-15 cell line was more sensitive (Appendix A). On the other hand, SW620 and HT-29 spheroids were not directly affected by ZA-SPNs nor by soluble ZA at the concentrations used, while the effect on HCT-15 spheroids was detected only with ZA (Appendix A). These findings were confirmed even after 96 h of incubation (not shown).

### 2.5. ZA-SPNs Elicited Vδ2 T Cell Proliferation from CRC Patients and the Killing of Autologous Tumor Organoids

To plan a potential therapeutic use of nanoformulated ZA, it is essential to prove that ZA-SPNs are actually able to trigger Vδ2 T cells from CRC patients’ samples. Toward this aim, first, the ability of ZA-SPNs to expand Vδ2 T cells, either from peripheral blood or from tumor biopsies of CRC patients, was assessed; then, the cytotoxic potential of these cells from patients’ PBMCs was quantified against CRC tumor cell lines or autologous tumor cells growing in vitro as colon mucosa organoids. A ZA-SPNs dose of 0.05 µM was chosen since it was the most effective on healthy donors’ PBMCs (see Figure 1A,B). Figure 5A shows that expansion of Vδ2 T cells from the PBMCs of CRC patients was obtained with 0.05 µM ZA-SPNs. This expansion occurred in all samples from CRC patients (n = 20), and in 14 out of 20 samples reached ≈90% after 14 days; however, a 20-fold higher concentration (1.0 µM) of soluble ZA relative to the ZA-SPN concentration was needed to achieve this effect (Figure 5A). The absolute Vδ2 T cell numbers are reported in Appendix A. Of note, ZA-SPNs were able to expand tumor-infiltrating Vδ2 T cells (Appendix A reports the pathological features of these patients) when added to mononuclear cells isolated from the tumor (Figure 5B); the percentage of Vδ2 T cells obtained ranged from 10 to 90% in 5 out of 10 cases. This result was quite remarkable since in the bioptic cell suspensions, the starting amount of Vδ2 T cells was mostly lower than 0.1% and the CD45^+^ leukocytes, containing infiltrating lymphocytes, were always less than 35% of the total cells (range 3–32% n = 10), as we reported elsewhere [21,44,45]. In addition, the presence of monocytes in these cell suspensions was almost negligible (range 0.02–0.5%, not shown) [21,46]. To define whether ZA-SPN-stimulated Vδ2 T cells from patients’ PBMCs belonged to a population of effector cells, we analyzed the expression of CD27 and CD45RA that was reported to define different lymphocyte subsets. In particular, naïve (N) T cells bear CD27 and CD45RA molecules, as well as central memory (CM) T lymphocytes; express only CD27; effector memory (EM) T cells are double negative; and terminal differentiated memory cells (TEMRA) are surface CD45RA positive [45,47]. We reported that ZA can drive the expansion of Vδ2 T cells, showing the characteristics of effector memory (EM) T lymphocytes (i.e., absence of CD27 and CD45RA) [48]. As shown in Appendix A, also ZA-SPNs-stimulated Vδ2 T lymphocytes displayed an EM/TEMRA (CD27^−^CD45RA^−^/CD27^−^CD45RA^+^) phenotype.

Thus, Vδ2 T effector cells obtained upon ZA-SPNs stimulation from the PBMCs of three CRC patients were tested for their anti-tumor cytotoxic activity against the CRC cell lines HCT-15 and HT-29, either as adherent monolayers (Figure 5C) or as spheroids (Figure 5D), exposed to ZA-SPNs. The E:T ratio chosen was again 3:1 to emphasize the effect due to the nanoformulated ZA taken up by tumor cells. Indeed, at this E:T ratio, the percentage of living tumor cells upon incubation with Vδ2 T cells was still more than 80% for HCT-15 and ≈70% for HT-29 (Figure 5C,D). When the target tumor cells or spheroids were challenged with 0.05 µM ZA-SPNs, the cytotoxic effect of Vδ2 T effector cells was much more evident, as living tumor cells were less than 50% in all instances, decreasing by up to 20% for adherent HT-29 or HCT-15 spheroids (Figure 5C,D).

To test whether ZA-SPNs-expanded Vδ2 effector T lymphocytes could kill tumor cells in an autologous setting, organoids were derived from specimens of five CRC patients (listed in Appendix A) in serum-free controlled culture conditions. Appendix A shows these organoids, with variable sizes (perimeters and areas in Appendix A), and expressing the epithelial marker EPCAM (Appendix A). In particular, OMCR18-006TK organoids display a morphology typical of mucinous tumors, according to the histopathological diagnosis (Appendix A). We first demonstrated the ability of organoids to engulf ZA-SPNs within 24 h of exposure (Appendix A). Then, organoids, labeled with the green fluorescent probe calcein-acetoxy methyl ester (AM) and seeded in Geltrex domes, were exposed to ZA-SPNs (0.05, 0.1, 0.5 µM) and cultured with the corresponding autologous ZA-SPN-expanded Vδ2 T lymphocytes for 6 days to allow the effector cells to enter the domes and reach the organoids. Figure 5E shows a representative co-culture with Vδ2 T cells invading some organoids, where one is evident in subpanel a, and one appears almost destroyed in b; this is also shown with Z-stack images in Appendix A. The cytotoxic effect of Vδ2 T cells was evaluated by measuring the decrease in green fluorescence, as previously reported by Chung et al. [49]. Autologous ZA-expanded Vδ2 T cells decreased the vitality of tumor organoids by 10–30%; this cytotoxic effect was further increased when ZA-SPNs were added during the cytolytic assay. Indeed, in the presence of 0.05 µM ZA-SPNs, Vδ2 T cells could reduce the organoid viability by 50% (Figure 5F). In turn, 10- to 50-fold higher concentrations of soluble ZA were needed to obtain superimposable effects with soluble ZA (Figure 5F). Altogether, these results are the experimental proof that ZA-SPNs can efficiently trigger the growth and function of Vδ2 anti-tumor effector T cells from CRC patients.

### 2.6. ZA-SPNs-Vδ2 T Lymphocytes Transendothelial Migration Toward CRC Tumor Cells in a Microfluidic System

To test the ability of ZA-SPN-expanded Vδ2 T cells (i.e., Vδ2 T lymphocytes obtained from PBMCs upon stimulation with ZA-SPNs and further culture for 21 days) to extravasate and reach CRC cells, a double-channel microfluidic chip was used [50]. This device comprised a vascular compartment (VC), uniformly coated with human umbilical vascular endothelial cells (HUVECs), and an extravascular compartment (EC), hosting a Matrigel matrix with CRC cells. The two compartments were separated by an array of micropillars, equally spaced by 3 μm (Figure 6A). HUVECs were seeded and also grew on the micropillars area, realizing an authentic vascular bed (Figure 6A,B showing the 3D reconstruction). By using this microfluidic system, the dynamics of ZA-SPN-expanded Vδ2 T cells infused into the VC was monitored over time. In the time-lapse experiments, these Vδ2 T cells fluxed into the VC (visible in blue due to the nuclear staining) and were able to cross HUVEC monolayer (red, CM-Dil membrane staining), and through the array of pillars, infiltrate the Matrigel matrix and reach the tumor cells (green, GFP^+^) (Figure 6C).

The fluorescence intensity of the Vδ2 T cell nuclear staining was measured after 12 and 24 h in the VC and EC using the Image J software, quantifying the percentage increase over time in different regions of interest (ROIs) set for each channel (Figure 6D). Extravascular Vδ2 T cell accumulation was detectable within 24 h, with a fluorescence increase of about 300% in EC (Figure 6D, black histograms). While transported by the flow into the VC, the fluorescence increased up to 140 and 190% within the first 12 and 24 h of observation, respectively, documenting the interaction of Vδ2 T cells with endothelial cells (HUVECs) during the migration process (Figure 6D, white histograms). This interaction was also visible in confocal images of a separate experiment, in which Vδ2 T cells (CM-Dil stained, red) appeared in part in contact with HUVEC (green, phalloidin staining F-actin) in the VC and in part adherent to the CRC in the EC (green, phalloidin staining F-actin); notably, CRC cells interacting with Vδ2 T lymphocytes displayed a dim fluorescence and an altered actin distribution (Figure 6E). In conclusion, Vδ2 T cells grown upon a ZA-SPN stimulation, and tested in the microfluidic system, displayed the ability to extravasate and interact with tumor cells in the EC.

## 3. Discussion

A major challenge in the hypothetical use of N-BPs as stimulators of anti-tumor Vδ2 T lymphocytes in solid cancers is related to the bone tropism of these compounds [16,17]. The encapsulation of ZA into spherical polymeric nanoparticles (ZA-SPNs), with a characteristic size of ≈200 nm, appears to be an effective solution to this hindrance. Indeed, nanoparticles up to a few hundreds of nanometers in diameter tend to passively accumulate in tumor tissues, following the so-called enhanced permeability and retention (EPR) effect [30,33,34]. This effect warrants the release of the drug into the tumor, preventing its early leakage and its natural accumulation in bone tissue.

In this work, we have demonstrated that SPNs can efficiently load ZA in such a way that large-scale production for clinical translation could be possible and that the payload was stably retained within its aqueous core for up to approximately one week. The architecture of ZA-SPNs, with an aqueous core confined within a lipid monolayer, resemble that of natural vesicles, likely favoring their penetration into cells. This would cause a progressive release of ZA inside tumor cells, leading to prompt activation of the mevalonate pathway responsible for IPP production [10], which eventually induces the activation and proliferation of Vδ2 T lymphocytes with anti-cancer functions.

Interestingly, not only do ZA-SPNs stimulate Vδ2 T cells at a concentration about 300-fold lower compared to soluble ZA, but can also, and unexpectedly, trigger their proliferation without exogenous IL-2, which is usually needed for T cells expansion. Possibly, ZA-SPNs induce a stronger signal than soluble ZA onto Vδ2 T cells, and in this way, promote the expression of activation molecules, such as the metabolic gatekeeper CD69 and the α-chain of the IL-2 receptor (CD25). Indeed, although not shown, a small fraction (≈17%) of Vδ2 T lymphocytes could actually express intracytoplasmic IL-2 upon stimulation with ZA-SPNs. Thus, ZA-SPNs could sensitize Vδ2 T cells to cytokines and growth factors present in the extracellular milieu. Yet, the underlying mechanism would have to be more precisely characterized in further studies.

Of note, ZA-SPNs were able to enhance the proliferation of Vδ2 T lymphocytes derived from the peripheral blood or tumor specimens of CRC patients. Based on the EC_50_, the efficiency of ZA-SPNs to expand Vδ2 T lymphocytes was 2 to 3 orders of magnitude higher than that of soluble ZA. These Vδ2 T lymphocytes behaved as effector cells, exerting a strong cytolytic activity against CRC tumor cells exposed to ZA-SPNs. This was tested in two different 3D models, namely tumor spheroids and patient-derived organoids, to mimic cancer cell growth in vivo more closely. Indeed, there is increasing evidence that both tumor development and response to therapy in humans are not always predictable in animals [51,52]. Recently, zoledronic-acid-containing nanoparticles have been reported to localize into extra-skeletal tumors in a murine model, thus avoiding bone sequestration of such drug formulation [53]. This would also reinforce the proposal to use ZA-SPNs in therapeutic schemes aimed to activate an anti-cancer immune response, mainly through the activation of γδ T lymphocytes. Regrettably, murine cancer models present several limitations, considering that the two major γδ T cell subsets do not have orthologues in mice [54,55], and that even syngeneic mice would not be adequate as the response to N-BPs of γδ T cells is different in rodents than in humans [6,8]. On the other hand, 3D culture systems, including spheroids and organoids, have been validated as preclinical models to overcome these inconveniences by the present authors and others [44,45,56,57].

In a first set of 3D experiments, spheroids with diameters ranging from 50 to 250 µm were considered to mimic tumors at early stages. In this case, ZA-SPNs could trigger Vδ2 T lymphocyte-mediated cytotoxicity with different efficiencies depending on the used cell line. It is of note that this difference among the used CRC cell lines was not evident on the cell monolayers. This may have been due to the spatial heterogeneity in the IPP production by CRC cells because of the diverse depths of penetration of the drug within the 3D cell aggregates. The cell line SW620 generates compact spheroids that, although capable of ZA-SPNs engulfment, may limit drug penetration compared to HCT-15 and HT-29 spheroids [45]. Another explanation is based on the different mevalonate metabolism and IPP production that follows ZA-SPNs’ entry; these features might be constitutive of each cell line or vary depending on the 3D structure acquired during tumor growth. Interestingly, the cytotoxic efficiency of ZA-SPNs was remarkable when patients’ organoids were exposed to autologous Vδ2 T cells that had expanded upon stimulation with ZA-SPNs. This demonstrated the potential therapeutic relevance of ZA-SPNs in CRC patients, suggesting that 3D culture systems may have specific features influencing the treatment outcome. Although validated using reliable pre-clinical models [44,56,57], both spheroids and organoids have some limitations, including the lack of different cellular components of the tumor microenvironment. It should be recalled that effector lymphocytes must exert their anti-tumor activities in a microenvironment populated by suppressive cells, such as mesenchymal stromal cells (MSC), regulatory T lymphocytes (Treg), and myeloid derived suppressor cells (MDSC) [46,58].

Nevertheless, the present authors and others have previously reported that microenvironment immunosuppression can be reverted using ZA or phosphoantigens, resulting in Vδ2 T cell activation [48,59]. It is still to be established whether ZA-SPNs can affect the regulatory functions of Treg and MDSC. However, it is likely that ZA-SPNs could influence the function of leukocytes since a remarkable Vδ2 T cell stimulation was observed in the presence of a few monocytes. This should most likely be ascribed to the enhancement of IPP production by these cells.

Another property of ZA-SPN stimulation was that of enhancing the ability of Vδ2 T cells to sense tumor cells. Specifically, in a microfluidic chip, ZA-SPN-expanded Vδ2 T cells were able to extravasate and migrate toward tumor cells, overcoming the vascular barrier and the extracellular matrix. These findings indicate that Vδ2 T cells, after being activated by ZA-SPNs, could reach tumor cells and recirculate, even if triggered away from the tumor site.

## 4. Materials and Methods

### 4.1. Reagents

Zoledronic acid (ZA) was purchased from Selleckchem (Aurogene, Rome, Italy). Poly(D,L-lactide-co-glycolic) acid (PLGA, 50:50, CarboxyTerminated, molecular weight ≈ 60 kDa) was purchased from Sigma-Aldrich (St. Louis, MO, USA). 1,2-dipalmitoyl-sn-glycero-3-phosphocholine (DPPC), 1,2-distearoyl-sn-glycero-3-phosphoethanolamine-N-[succinyl(polyethylene glycol)-2000] (DSPE-PEG), and 1,2-distearoyl-sn-glycero-3-phosphoethanolamine (DSPE) were obtained from AvantiPolar Lipids (Alabaster, AL, USA). Analytical grade dimethyl sulfoxide (DMSO), acetonitrile, chloroform, absolute ethanol, dipotassium hydrogen phosphate anhydrous, and tetra butyl ammonium Bi-sulphateand 5(6) carboxyfluoresceindiacetate *N*-succinimidylester (CFSE) were purchased from Sigma-Aldrich (Milan, Italy). RPMI-1640 and FBS (One Shot™ Fetal Bovine Serum) were obtained from Gibco (Thermo Fisher Scientific, Monza, Italy). DMEM-F12,L-Glutamine, and penicillin/streptomycin (BioWhittaker^®^ Reagents) were purchased from Lonza (Basel, Switzerland). Epidermal growth factor (EGF) was purchased from Peprotech Europe (London, UK) while IL-2 was purchased from Miltenyi (Miltenyi Biotec Italia, Bologna).

### 4.2. Zoledronic Acid-Loaded Spherical Polymeric Nanoparticles (ZA-SPNs) Synthesis

Spherical polymeric nanoparticles (SPNs) were obtained using a double emulsion–evaporation procedure, modifying the single emulsion protocol previously described by Lee et al. [38]. Briefly, ZA, dissolved in the aqueous phase, was slowly added under probe sonication to an organic phase containing 10 mg of PLGA and DPPC dissolved in chloroform. This first emulsion was added dropwise to a 4% ethanol solution containing DSPE-PEG under probe sonication. The molar ratio of DPPC:DSPE-PEG was 7.5:2.5, while both the lipids were at 20% *w*/*w* of the polymer. Empty SPNs were prepared following the same procedure, but without ZA in the aqueous phase. Fluorescent ZA-SPNs were prepared by substituting a small fraction of DPPC (10% of the total amount) with DSPE-Cy5. After the evaporation of all the organic solvent in a reduced pressure environment, SPNs were purified and collected through sequential centrifugation steps. The first centrifugation was performed at 1200 rpm for 2 min to remove large debris from the synthesis process. The supernatant was then centrifuged at 12,000 rpm for 15 min, and the remaining pellet was centrifuged at the same speed several times in order to remove the ZA not incorporated into the SPNs. Finally, the resulting SPNs were resuspended in 1 mL aqueous solution before their use in all the subsequent experiments.

### 4.3. ZA-SPNs Physico-Chemical and Pharmacological Characterization

The nanoparticle size distribution and PDI were measured at 37 °C using dynamic light scattering (DLS) with the Zetasizer Nano ZS (Malvern, UK). By using proper zeta-cells, the nanoparticles’ ζ-potential was also measured. For the stability study, both the size and PDI were measured over time for a period of 2 weeks while maintaining nanoparticles at 37 °C in deionized (DI) water. Also, ζ-potential was measured and monitored for the same time period. To study the nanoparticle morphology, SPN samples were dropped on a silicon wafer and dried. Samples were then gold sputtered and analyzed using a JSM-7500FA (JEOL, Milan, Italy) analytical field-emission scanning electron microscope (SEM) at 15 keV. The amount of ZA entrapped in the nanoparticles (n = 3 for each experimental condition) were measured using HPLC (1260 Infinity, Agilent Technology, Milano, Italy), using a reverse phase C-18 column (Zorbax Eclipse plus, Agilent Technology, Milano, Italy). Samples were eluted in isocratic conditions using a mixture of methanol (5%), acetonitrile (12%), and a buffer made out of 4.5 g of dipotassium hydrogen phosphate anhydrous plus 2 g of tetra butyl ammonium bi-sulphate in 1 L of DI water. The provided molarity refers to the molarity of one batch of ZA-SPNs resuspended in 1 mL of solution. To evaluate the release profile of ZA from the nanoparticles, a known amount of ZA-SPNs was loaded into Slide-A-Lyzer MINI dialysis microtubes with a molecular cut-off of 10 kDa (Thermo Fisher Scientific, Waltham, MA, USA), and placed in 4 L of PBS in order to simulate the infinite sink condition. At predetermined time points (namely 1, 4, 24, 48, 72, 112, and 158 h), three samples were collected and the amount of ZA was measured using high pressure liquid chromatography (HPLC).

### 4.4. Patients

Twenty-six CRC patients suffering from CRC were studied (institutional informed consent signed at the time of donation and EC approval PR163REG201 renewed in 2017). The localization of tumors was determined by the surgery staff of the Oncological Surgery Unit of the Istituto di Ricerca e Cura a Carattere Scientifico (IRCCS) Ospedale Policlinico San Martino. The tumor stage was determined according to the Union for International Cancer Control (UICC) and Dukes classification modified by Aster and Coller [60], and the microsatellite status was analyzed by the Pathology Unit. The PBMCs were isolated from all patients and used for measuring Vδ2 T lymphocyte proliferation and cytotoxic activity in an allogenic or autologous setting. Tumor specimens from 14 patients were analyzed (Appendix A): 10 for the isolation of cell suspensions, used in experiments aimed to determine the ability of ZA-SNPs to trigger the expansion of Vδ2 T cells, and 5 for the generation of organoids and used, within the fourth passage of culture, as targets to evaluate the cytotoxic activity of Vδ2 T cells from autologous PBMCs.

### 4.5. Ex Vivo Expansion of Vδ2 T Cells

ZA was solubilized in DMSO, following the manufacturer’s instructions. The amount of soluble ZA to trigger Vδ2 T cell proliferation or activation of Vδ2-T-cell-mediated tumor cell lysis ranged from 0.5 μM to 5 μM, in keeping with our previous data [21,45,48]. At these concentrations, the dilution of DMSO in culture was less than 1:10^3^ (between 1:2 × 10^3^ and 1:2 × 10^4^). Neither DMSO at 1:10^3^ nor ZA at concentrations up to 1 µM induced toxic effects, as evaluated using a crystal violet assay and propidium iodide (PI, Sigma-Aldrich) staining, on the cells used in this study, nor did they influence the proliferation or cytotoxicity of Vδ2 T lymphocytes [45]. Using ZA at a 5-µM concentration, about 20% of dead cells were detected among the Vδ2 T cells after 14 days of culture. The amount of ZA-SPNs to be used for triggering the expansion of Vδ2 T cells was determined in preliminary experiments by adding decreasing amounts of SPNs to cell cultures (namely 0.5, 0.25, 0.12, 0.05, 5 × 10^−3^, 5 × 10^−4^, 5 × 10^−5^ µM). Notably, in all the in vitro experiments, the maximum concentration used of ZA-SPNs was 0.5 μM and the optimum was 0.05 μM, the latter being more than a thousand times lower than the batch concentration and free of direct toxic effect on T lymphocytes. Similarly, this protocol was used also to estimate the non-toxic amount of nanoparticles for PBMCs (not shown). As a control, empty nanoparticles were used, with the same polymer amounts used for the ZA-SPNs. A concentration of 0.5 µM for 10^5^ cells, the highest concentration tested, was not toxic for either lymphocytes nor monocytes. A concentration of 50 nM of ZA-SPNs (from now on 0.05 µM for easier comparison with soluble ZA) was experimentally found to be adequate to follow the expansion of Vδ2 T cells in in vitro cultures.

PBMCs were obtained from both healthy adult donor’s buffy coats (institutional informed consent signed at the time of donation) and venous blood samples of CRC patients using density gradient centrifugation with Lymphocyte Separating Medium (Pancoll human, density: 1.077 g/mL, PAN-Biotech, Munich, Germany), as described in Zocchi et al. [21]. To obtain Vδ2 T lymphocyte populations, PBMCs were cultured in 96-well U-bottomed plates in 200 μL of RPMI-1640 medium supplemented with 10% FBS, penicillin/streptomycin, and L-glutamine, and with serial dilutions of either free ZA or ZA-SPNs, in a 37 °C humidified cell incubator with 5% CO_2_.

After 24 h, and on days 5 and 7, 100 µL of culture supernatant were discarded and substituted with 100 µL of fresh medium containing recombinant human IL-2 (30 UI/10 ng/mL final concentration, Miltenyi Biotec Italia, Bologna). On day 10, cells were split in the medium supplemented with IL-2 and this was repeated every three days. The percentage of Vδ2 T lymphocytes was determined at different time points (days 0, 7, 14, 21) using indirect immunofluorescence and cytofluorimetric analysis, which was always performed by gating viable cells using the anti-Vδ2 TCR specific monoclonal antibody (mAb) γδ123R3, as described in Musso et al. [48]. Lymphocyte populations were used as effector cells in co-culture experiments with tumor cells or tumor cell spheroids after day 21 when the percentage of Vδ2 lymphocytes were more than 96% of the total cells. In some experiments, T lymphocytes and monocytes (Mo) were isolated from PBMC using specific negative selection kits (Stemcells Biotecnologies, Voden, Italy). The purity of the selected T lymphocyte and Mo populations was always more than 97% and 95%, respectively; this was determined using immunofluorescence with specific anti-CD3 (to identify T cells) or anti-CD14 (to stain Mo) mAbs and FACS analysis (see below). Then, T cells and Mo were co-cultured in the presence of soluble ZA or ZA-SPNs at different T:Mo ratios (10:1, 20:1, 40:1, 100:1, 200:1, 1000:1) and the percentage of Vδ2 cells was analyzed using immunofluorescence with the specific anti-Vδ2 mAb.

### 4.6. Immunofluorescence Assay and Analysis of Lymphocyte Proliferation

The immunofluorescence assay was performed as described in Musso et al. [48] with anti-Vδ2 mAb (γδ123R3, IgG1) or anti-CD3 mAb (289/10/F11, IgG2a) or anti-CD69 (31C4, IgG2a) or anti-CD25 (4E3, IgG2b, Miltenyi Biotec) or anti-CD45RA (T60, IgG2a) or anti-CD27 (MT271, IgG1, Miltenyi Biotec), anti-EPCAM (15806, IgG2b, R and D System, Minneapolis, MN) or anti-CD14 (TUK4, IgG2a, Thermo Fisher Scientific) mAbs, followed by Alexafluor 647 or PE-anti-isotype specific goat anti-mouse antiserum (GAM) (Life Technologies, Milan, Italy). At least 10^4^ cells/sample were run on a CyAn ADP cytofluorimeter (Beckman-Coulter Italia, Milan, Italy), and results were analyzed with the Summit 4.3 software (Beckman-Coulter) and expressed as percentage of fluorescent cells or mean fluorescence intensity arbitrary units (MFI a.u.).

To measure the proliferation of Vδ2 T lymphocytes, the PBMCs were labelled with CFSE as described in Musso et al. [61]. Briefly, 10^6^ cells were incubated for 30 min at 37 °C in a water bath in complete medium with 100 nM CFSE. Then, cells were extensively washed and put in culture at 10^5^ cells/microwell in 96-well U-bottomed plates. ZA (1 μM) or ZA-SPNs (0.05 μM) were added and the proliferation was analyzed at 7 days after labelling cells with anti-Vδ2-specific mAb, followed by Alexafluor 647 GAM. Samples were analyzed on a CyAn ADP cytofluorimeter and proliferation was indicated by the reduction of CFSE in the cell generations compared to the content of CFSE in the parental component. The different cell generations were defined using the software ModFit LT 5.4 (Verity Software House, Topsham, ME, USA) with different colors.

### 4.7. CRC Cell Cultures and Spheroid Generation

The human CRC cell lines SW620, HCT-15, HT-29, LS180, SW-48, and SW480, provided and certified as mycoplasma-free by the cell bank of the IRCCS Ospedale Policlinico San Martino (Blood Transfusion Centre, B. Parodi, Genoa, Italy), were cultured in RPMI-1640 medium supplemented with 10% FBS, penicillin/streptomycin, and L-glutamine in adherent culture plates in a humidified incubator at 37 °C with 5% CO_2_. Experimental conditions for the generation of tumor cell spheroids were selected starting with decreasing numbers of each tumor cell line (2 × 10^4^ to 1 × 10^4^ to 5 × 10^3^ per well) in flat-bottom 96-well plates (Ultra-Low attachment multiwell plates, Corning^®^Costar^®^, New York City, NY, USA), with DMEM-F12 serum free medium supplemented with EGF (Peprotech Europe, London, UK) at a 10 ng/mL final concentration (≥1 × 10^6^ IU/mg). EGF was selected, among other natural ligands of epidermal growth factor receptor (EGFR), due to reasons previously described in detail in Varesano et al. [45]. When plating, 10^4^ cells of SW620, HCT-15, or HT-29 tumor cell spheroids were obtained in 5–6 days after changing the culture medium every two days [45].

On day 5, CRC spheroids with a maximum diameter of about 250 μm were composed of living cells, as assessed by culturing a sample under adherent conventional conditions for 24 h and the subsequent identification of living cells with propidium iodide (PI, Sigma-Aldrich) staining and a crystal violet assay (data not shown) as described in Varesano et al. [45]. Spheroid dimensions were measured using images taken with the Olympus IX70 bright field inverted microscope equipped with a CCD camera (ORCA-ER, C4742-80-12AG, Hamamatsu, Japan) via the analysis of regions of interest after defining each spheroid with the CellSens software (version 1.12, Olympus, Tokyo, Japan) [43].

### 4.8. CRC Cell Viability and Cytotoxicity Assay

The CRC cell viability was determined using the Crystal Violet Cell Cytotoxicity Assay Kit (Biovision, Milpitas, CA, USA) either upon exposure to serial dilution of ZA-SPNs from 0.5 to 0.005 µM or ZA from 5 µM to 0.5 µM, and/or following co-culture with Vδ2 T cells. In conventional adherent cultures, CRC cells were incubated with ZA-SPNs or free ZA at the above-mentioned concentrations with Vδ2 T cells for 48 h, while finding the minimum time point to detect cytotoxicity with this system [45]. The optimal amount of Vδ2 T cells to detect the cytotoxic effect elicited by the drugs added, as determined in preliminary experiments, was 7.5 × 10^4^ Vδ2 T cells/2.5 × 10^4^ CRC cells, corresponding to a 3:1 effector to target (E:T) ratio. Similarly, CRC spheroids were exposed to ZA-SPNs or ZA, as above, and co-cultured for 48 h with Vδ2 T cells at the E:T ratio of 3:1 calculated as described in Varesano et al. [45]. Then, spheroids were transferred in conventional adherent plates and incubated for an additional 24 h to allow for plastic attachment. The CRC cells were then washed four times with PBS and the adherent cells were stained with crystal violet following the manufacturer’s instructions. The amount of crystal violet, proportional to the amount of adherent/living cells, was measured with the Victor X5 multilabel plate reader (Perkin Elmer Italia SPA, Milan, Italy) at 595 nm [45]. Results are expressed as a percentage of living cells calculated as follows: (OD_595_ CRC/OD_595_ CRC plus drug and/or Vδ2 T cells) × 100%.

### 4.9. CRC Organoid Generation and Vδ2 T Cell Cytotoxicity Elicited Using ZA-SPNs

Primary CRC organoid cultures were obtained following published guidelines [62,63]. Tissue samples, obtained after patient informed consent (PR163REG2014 renewed in 2017) and collected by a trained pathologist, were enzymatically digested by collagenases type I and II in Leibovitz L15 medium (Gibco-Thermo Scientific Italia, Milan, Italy) without serum. The digested tissue was passed through a 100-μm strainer to completely eliminate the residual matrix and mucus. Cripts were washed several times in fresh L15 medium to eliminate cell debris, mixed with Geltrex (LDEV-free, hESC-qualified, reduced growth factor; Gibco-Thermo Scientific), and plated in a 24-well plate. After polymerization at 37 °C, Geltrex domes were covered with 500 μL of medium per well (DMEM-F12 plus B27 plus 10 ng/mL of EGF, ≥10^6^ IU/mg) supplemented with antibiotics. Different cocktails of inhibitors were tested on different wells [61] to identify the best culture condition for each CRC tumor sample. This culture method naturally selects pure colorectal epithelial cells within a few in vitro passages, while not allowing for the expansion of other contaminating populations [64]. The absence of wnt3a and R-spondin in the culture medium excluded the contamination of normal epithelial cells from mucosa. In this study, we tested five organoid cultures (OMCR18-006TK, OMCR18-006TK, OMCR19-006TK, OMCR19-009TK, and OMCR19-016TK), as detailed in Appendix A, which were photographed using a Leica DM-LB2 microscope (Leica Biosystems, Milan, Italy), equipped with a GX-CamU3-18 camera (GT-Vision, Stansfield, UK) and depicted in Appendix A.

Organoids in Geltrex domes (3 µL) were labeled with calcein-AM (Sigma-Aldrich, 500 nM), washed with culture medium, seeded in 96-well flat-bottomed plates (1 dome/well), and challenged with autologous Vδ2 T cells, previously expanded with ZA-SPNs, at the E:T ratio of 3:1. The number of tumor cells in the 3-µL dome was evaluated via counting with the Miltenyi MACS Quant Cytofluorometer (Miltenyi Biotec srl, Bologna, Italy). The E:T co-culture was performed in the presence of ZA-SPNs (0.5, 0.1, 0.05 µM) and prolonged for 6 days to allow for the penetration of lymphocytes into the domes (3–4 days, as evaluated using conventional microscopy in preliminary experiments, not shown), followed by effector cell function. Then, fluorescence of each culture well was quantified using spectrofluorometry (Ex:488 nm, Em: 530 nm, Victor X5, Perkin Elmer) and compared to the fluorescence of organoid cultures without Vδ2 T cells, which was considered to be 100%. Data are expressed as a percentage of green fluorescence.

### 4.10. Cell Preparation for Confocal and SEM Imaging

SW-48 cells were seeded on circular glass slides previously coated with fibronectin (1 mg/mL, Sigma-Aldrich). After 24 h at 37 °C, to allow for cell attachment on the slide, samples were incubated with 0.05 µM Cy5-conjugated ZA-SPNs for 4 h. Vδ2 T cells were then added for an additional 4 h. For confocal imaging, cells were washed three times with PBS, fixed with 4% paraformaldehyde, and stained with DAPI and anti-EPCAM mAb, followed by Alexafluor555 GAM for CRC cells and with the anti-Vδ2 specific mAb or Vδ2 followed by Alexafluor488 GAM. Images were taken with a spinning disc confocal microscopy system (Eclipse Ti with Revolution XDi acquisition System, Nikon Instruments, Firenze, Italy) in sequence mode to avoid overlapping among the different fluorochromes. In some experiments, SW620 spheroids, the OMCR18-016TK organoids, or Vδ2 T cells obtained from the PBMCs were exposed to 0.05 µM Cy5-ZA-SPNs for 24 h. Spheroids and T cells were also stained with 20 nM Syto16 (Life Technologies, Thermo Fisher Scientific, Milan, Italy) to identify nuclei (blue pseudocolor). Spheroids or Vδ2 T cells were seeded onto glass slides, while organoids were analyzed in Geltrex domes seeded into 96-well black plates with a clear bottom (Costar, Corning. Inc., Minneapolis, MN, USA). In another series of experiments, ZA-SPN-expanded Vδ2 T cells were labeled with CFSE and incubated with the OMCR18-016TK organoids for 48 h and then analyzed using confocal microscopy. Samples were run under a FV500 confocal microscope (Olympus Italia srl, Milan, Italy with a PlanApo 20× objective or 40× NA1.00 oil objective, and data was analyzed with FluoView 4.3b software (Olympus Italia srl, Milan, Italy). Images were taken in sequence mode and shown in pseudocolor. For SEM, samples were washed with PBS and fixed with glutaraldehyde 2% in a sodium cacodylate buffer 0.1 M at pH 7.4; then, samples were post-fixed with osmium tetroxide 1% solution, dehydrated with a series of alcohol at 4 °C, and infiltrated with hexamethyldisilazane. After an overnight drying, they were sputter-coated with a thin (10 nm) layer of gold to protect them and make their surface conductive. Imaging was performed with an analytical low-vacuum SEM JSM 6490 (JEOL (Italia), Milan, Italy), at 15 kV.

### 4.11. Microfluidic Chip Fabrication

The double-channel microfluidic chip was realized following the same protocols of the present authors’ recent paper [50]. Briefly, by using a direct laser writing machine (DWL 66^+^, Heidelberg Instruments, Heidelberg, Germany), optical masks were obtained and then used in sequential photolithographic processes to impress the two channels and the raw pillars, which separated them on the resist spun on a silicon wafer. After the resists development, the entire 2D geometry was made tridimensional through the inductively coupled plasma – reactive ion etching (ICP-RIE) that, using a Bosh process, dug the pattern into the silicon. This entire design was then transferred to polydimethylsiloxane (PDMS), the final material of the chip, by casting the latter onto the silicon and baking it at 60 °C overnight. Finally, the PDMS was treated with an oxygen plasma (20 W for 20 s) and bonded to a glass coverslip to produce a closed hollow structure. A biopsy punch was used to create the inlets and outlets. The final microfluidic chip presented two channels, which were 210 μm wide and 50 μm high. The region of contact of the two channels (500 μm in length) was constituted by an array of pillars, separated by a 3-μm gap.

### 4.12. Microfluidic Experiments

Before the seeding of the cells in the channels, the chip was autoclaved at 120 °C. A solution of Matrigel (8–12 mg/mL, Sigma-Aldrich) was half diluted with a suspension of the CRC cell lines SW-48 or SW-480 GFP^+^ (kindly provided by Dr. N.Ferrari, 15 × 10^6^ cells/mL) and inserted in the extravascular channel. For the gelation of Matrigel, the chip was kept in an incubator for 5 min at 37 °C. Then, fibronectin (20 μg/mL, Sigma-Aldrich) was pipetted into the vascular channel, and after 15 min, HUVECs (6 × 10^6^ cells/mL, PromoCell GmbH, Heidelberg, Germany) were inserted into the same channel. Experiments were then performed after overnight incubation in order to get an endothelial vessel in the vascular channel and to allow tumor cells to attach and adapt to the extravascular channel environment. Vδ2 T cells, previously stained with DAPI or CM-(Invitrogen Thermo Fisher Scientific, Milan, Italy) were injected into the vascular channel through a syringe pump (Harvard Pump 11 Elite, Harvard Apparatus, Holliston,, MA, USA) at a flow rate of 50 nL/min and at a E:T (Vδ2:CRC) ratio of 3:1. A Spinning disc confocal microscopy system (Nikon EclipseTi with Revolution XDi acquisition System) acquired images at predefined time intervals (4 min) for 24 h. Images were analyzed using the ImageJ software (1.52n, NIH, USA), quantifying the percentage increase in fluorescence intensity (i.e., DAPI for Vδ2 T cells) in different regions of interest (ROIs, 500 μm in length and 215 μm in height), close to the pillar area. Data represent the percentage increase of such fluorescence, compared to time 0, and are expressed as mean ± SE. For the confocal images, F-actin cytoskeleton filaments were stained in green using phalloidin (Alexa Fluor phalloidin, Life Technologies, Thermo Fisher Scientific, Milan, Italy), nuclei with DAPI and Vδ2 T cells with CM-Dil. Images were acquired using a confocal microscope (Nikon A1).

### 4.13. HPLC Negative-Ion Electrospray Ionization TOF-MS

The production of IPP by monocytes or LS180 or SW620CRC cells, after treatment with different amounts of 0.5 or 0.05 µM ZA-SPNs or 0.5 µM soluble ZA for 24 h, was evaluated on cell extracts, dissolved via vortex mixing in 80 µL MilliQ water and 250 µL Na_3_VO_4_, and cleared using centrifugation in an Eppendorf Minifuge (Eppendorf srl, Milan, Italy) (13,000 rpm, 3 min) using HPLC negative-ion electrospray ionization time of flight mass spectrometry (HPLC/TOF-MS) according to the method described by Jauhiainen et al. [65] with some modifications reported in detail in Zocchi et al. [21]. Calibration curves for IPP were generated with standards diluted in MilliQ water/0.25 mM Na_3_VO_4_ in the range 0.1–15 µM. The IPP content was determined using HPLC/TOF-MS, operating in reflection negative ion mode, using an Agilent 1200 series chromatographic system, equipped with G1379B degasser, G1376A capillary pump, and G1377A autosampler. Negative full-scan mass spectra were recorded using the Agilent’s Mass Hunter software version n. B.05.00 in the mass range of m/z 60–500. The full scan data were processed using the Agilent Mass Hunter Qualitative Analysis, ver. B.02.00 software. The amount of IPP was measured using the extracted ion current (EIC) peak area (EIC m/z 244.99 [M-H]^−^). Results are shown as pmol of IPP extracted using acetonitrile/total protein content in each cell lysate. The chromatographic method used could not separate the isomers IPP and 3,3-dimethylallyl pyrophosphate (DMAPP). The identity of the parent ion present in our cell extracts was confirmed by verifying the formation of fragment ions (m/z 79, m/z 159, m/z 177, and m/z 227), generating negative MS/MS spectra with a mass spectrometer Agilent 1100 series LC/MSD Trap, equipped with an orthogonal geometry electrospray source and ion trap analyzer (not shown) [21].

### 4.14. Statistical Analysis

Statistical analysis was performed using a two-tailed unpaired Student’s *t-*test. For the time-lapse confocal studies, at first for assessing the homogeneity of variances, the equal-variance assumption was tested using the Brown–Forsy test. ANOVA was performed to evaluate the differences between groups, followed by the Tukey-HSD post-hoc test. The *p*-values are shown in the text or in the figure legends. Results are shown as mean ± SEM or mean ± SD.

### 4.15. Human Subjects

All human studies described in this manuscript have been approved by the appropriate institutional review board(s) as indicated in the Ethic Committee approval PR163REG2014 renewed in 2017. The written informed consent was received from participants prior to inclusion in the study and it is stored in the Molecular Oncology and Angiogenesis Unit.

## 5. Conclusions

In conclusion, this study demonstrates, for the first time to the present authors’ knowledge, that the encapsulation of ZA into spherical polymeric nanoparticles can lead to the stimulation of Vδ2 T cell expansion and activation of anti-tumor activity far more effectively than soluble ZA. Moreover, the awareness that both peripheral-blood and tumor-infiltrating Vδ2 T cells can respond to ZA-SPNs and efficiently induce tumor cell death in patients’ organoids in an autologous setting, would further support the use of this ZA nanoformulation in CRC. The eventual therapeutic plan may be preceded by testing the IPP production and Vδ2 T cell activation in the patient-specific 3D model of tumor organoids in order to pursue a personalized therapy. These results are a step toward the use of ZA-SPNs to trigger anti-tumor immunity in CRC and their possible translation in clinical practice.

## Figures and Tables

**Figure 1 cancers-12-00104-f001:**
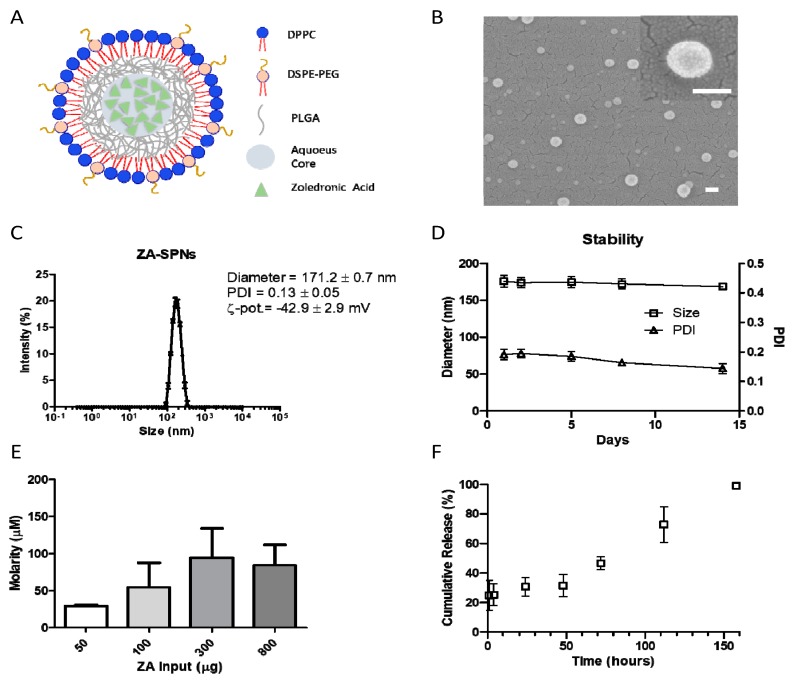
Zoledronic acid–spherical polymeric nanoparticles’ (ZA-SPNs) physico-chemical properties. (**A**) Schematic of a ZA-SPN. DPPC: 1,2-dipalmitoyl-sn-glycero-3-phosphocholine; DSPE-PEG: 1,2-distearoyl-sn-glycero-3-phosphoethanolamine-N-[succinyl(polyethylene glycol)-2000]; PLGA: Poly(D,L-lactide-co-glycolic) acid. (**B**) SEM images of ZA-SPNs, showing their spherical shape and dimension (enlargement in the upper-right corner). Scale bar: 100 nm. (**C**) Size distribution of ZA-SPNs; polydispersity index (PDI) and ζ-potential values are also shown. (**D**) Size and PDI stability over time of ZA-SPNs. (**E**) ZA-SPNs batch average molarity (µM) obtained for different input amounts of ZA in the synthesis process, showing 300 μg as the best input condition. (**F**) Release profile of ZA from ZA-SPNs, showing the minimal ZA loss within the first 48 h. Panels C–F: mean values ± SD.

**Figure 2 cancers-12-00104-f002:**
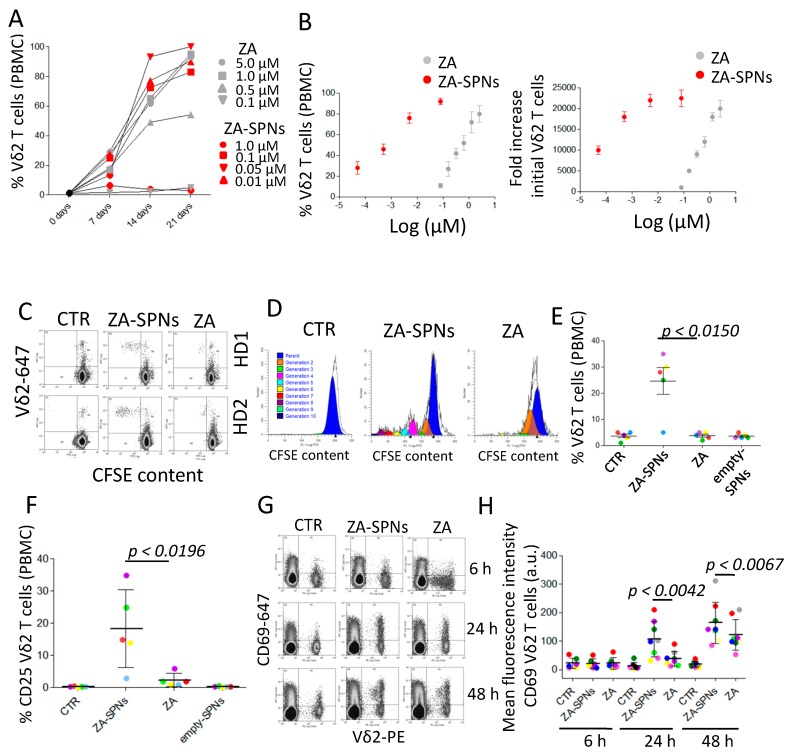
ZA-SPNs stimulated the expansion and activation of Vδ2 T lymphocytes. (**A**) PBMCs were incubated with ZA-SPNs (red) or soluble ZA (gray) at the indicated concentrations (μM) for 24 h before the addition of interleukin (IL)-2. The percentage of Vδ2 T lymphocytes was evaluated at different time points (7, 14, and 21 days) using fluorescence activated cell sorter FACS analysis. One representative experiment out of 10 is shown. (**B**) Left graph: PBMCs were incubated with ZA-SPNs (red) or soluble ZA (gray) (Log μM) for 24 h before the addition of IL-2. The percentage of Vδ2 T lymphocytes (2.5 ± 1.2% in the starting PBMCs) was evaluated on day 21 using FACS analysis. Mean ± SD of 10 healthy PBMCs. Right graph: Results of the left graph expressed as a fold increase of Vδ2 T lymphocytes/mL at the end of the culture compared to the initial number. Mean ± SD of 10 PBMCs. (**C**) Carboxyfluorescein succinimidyl ester (CFSE)-labeled PBMC from two representative donors (HD1, HD2) were cultured without (CTR) or with ZA-SPNs (0.05 μM) or soluble ZA (1 μM), without the addition of exogenous IL-2, and the decrease of CFSE intensity (green) was evaluated on day 7. Vδ2 T cells were identified with the specific mAb (red). (**D**) Vδ2 T cell proliferation, elicited as in (C), evaluated using the CFSE decrease in gated Vδ2 T cells with the ModFit LT5.4 software. CTR: CFSE content in cells with the medium alone. Experimental representative of five replicates performed. (**E**) Vδ2 T cell proliferation to ZA-SPNs or ZA or empty SPNs (same nanoparticle amounts as for the ZA-SPNs) elicited as in (C) in five donors; mean ± SD also shown. CTR: medium alone. (**F**) Expression of the IL-2Rα chain (CD25) at 48 h using FACS analysis on Vδ2 T lymphocytes in the indicated experimental conditions (CTR: medium alone; ZA-SPNs: 0.05 µM; ZA: 1.0 µM; empty SPNs). Results are expressed as a percentage of CD25^+^ cells in gated Vδ2 T lymphocytes. (**G**,**H**) PBMCs cultured as in (C) for 6 h, 24 h, or 48 h were stained with the anti-Vδ2 and anti-CD69 mAbs and analyzed using FACS. Quadrants: lower-left—CD69^−^Vδ2^−^ cells; upper-left—CD69^+^ cells; upper-right—CD69^+^Vδ2^+^ cells; lower-left—Vδ2^+^ cells. In (H), PBMCs stained as in (G) and evaluated as the mean fluorescence intensity (MFI, a.u.) of CD69 expression on Vδ2 T cells at 6 h, 24 h, and 48 h. Mean ± SD from eight donors’ PBMCs. CTR: medium alone.

**Figure 3 cancers-12-00104-f003:**
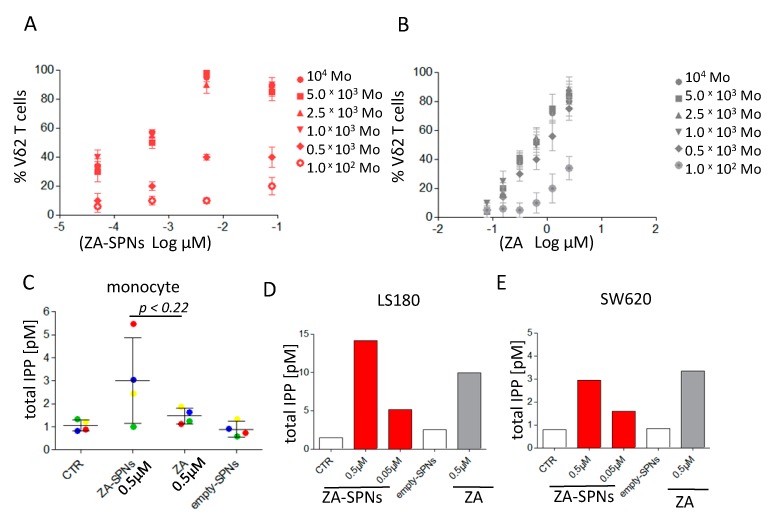
Monocyte requirement for Vδ2 T cell expansion and isopentenyl pyrophosphate (IPP) production in response to ZA-SPNs or ZA. (**A**,**B**) A total of 10^5^ purified T lymphocytes were cultured with different amounts of autologous purified monocytes (Mo), as indicated, in the presence of ZA-SPNs (A) or soluble ZA (B) at different concentrations for 21 days. The percentage of Vδ2 T cells present in culture was evaluated using indirect immunofluorescence and FACS analysis; results are the mean ± SD of T cell populations from six healthy donors. (**C**) A total of 10^6^ Mo were incubated for 24 h without (CTR) or with 0.5 μM ZA-SPNs or with 0.5 μM soluble ZA or with empty SPNs, and the amount of IPP produced (pM) was evaluated using high performance liquid chromatography/time of flight-mass spectrometry HPLC/TOF-MS. Colors indicate the four Mo donors. The mean ± SD for each condition is also shown. (**D**,**E**) A total of 5 × 10^5^ LS180 or SW620 CRC cells were incubated for 24 h with 0.05 or 0.5 μM ZA-SPNs or 0.5 μM soluble ZA, and IPP production was evaluated as in panel (C). Results are shown as pmol of IPP extracted by acetonitrile/total protein content in each cell lysate analyzed using HPLC/TOF-MS. One representative experiment of two performed is shown.

**Figure 4 cancers-12-00104-f004:**
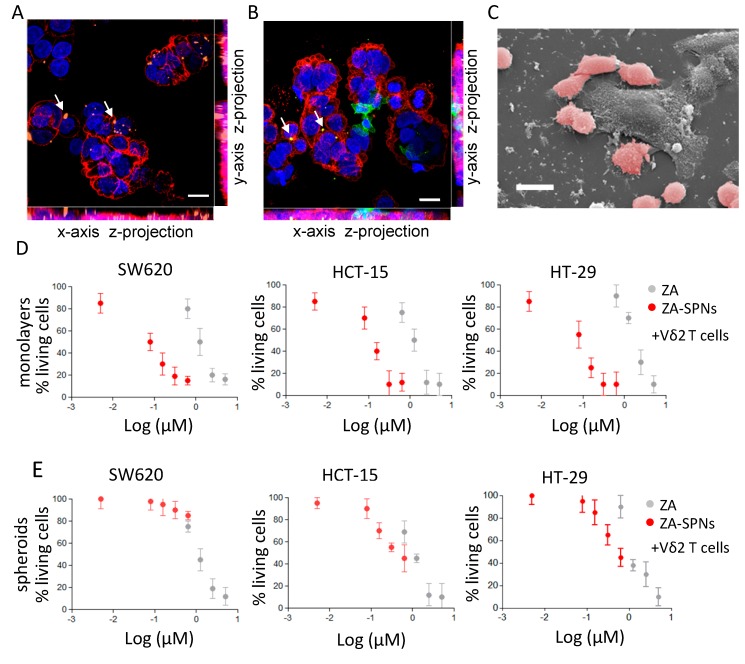
ZA-SPNs triggered the cytotoxicity of colorectal carcinoma (CRC) cell lines and spheroids using Vδ2 T cells. (**A**) Representative confocal microscopy image showing Cy5-conjugated ZA-SPNs’ (orange, arrows) internalization in CRC cells, identified using the anti-EPCAM mAb and Alexafluor555 isotype-specific goat antimouse (GAM) (red). Blue: nuclei diamidino phenylindole (DAPI) staining. Scale bar: 10 µm. (**B**) Vδ2 T cells identified using the specific anti-Vδ2 mAb and Alexafluor488-anti-isotype GAM (green) interacting with CRC (EPCAM followed by Alexafluor555 GAM, red) previously exposed (24 h) to ZA-SPNs (white arrows). Blue: DAPI nuclear staining. Scale bar: 10 µm. (**C**) SEM images of a similar experiment in which Vδ2 T cells (pink) had surrounded a CRC (gray). Scale bar: 20 µm. (**D**) The CRC cell lines SW620, HCT-15, and HT-29 were cultured adherent in flat bottomed plates with ZA-SPNs (red) or with soluble ZA (gray) at different concentrations and challenged with activated Vδ2 T cells from healthy donors at the effector:target (E:T) ratio of 3:1 for 48 h. Then, viability was assessed using a crystal violet assay. Data is expressed as a percentage of living cells as compared to cells cultured in medium alone, are the mean ± SD of three independent experiments of six different replicates for each condition. (**E**) Spheroids of the CRC cell lines SW620, HCT-15, and HT-29 at day 5 were incubated with activated Vδ2 T cells at the E:T ratio of 3:1 and either ZA-SPNs (red) or soluble ZA (gray) at the indicated concentrations for 48 h. Spheroid cell viability was assessed using crystal violet assay modified as described in Varesano et al. [45]. Data are the mean ± SD of three independent experiments of six replicates for each condition.

**Figure 5 cancers-12-00104-f005:**
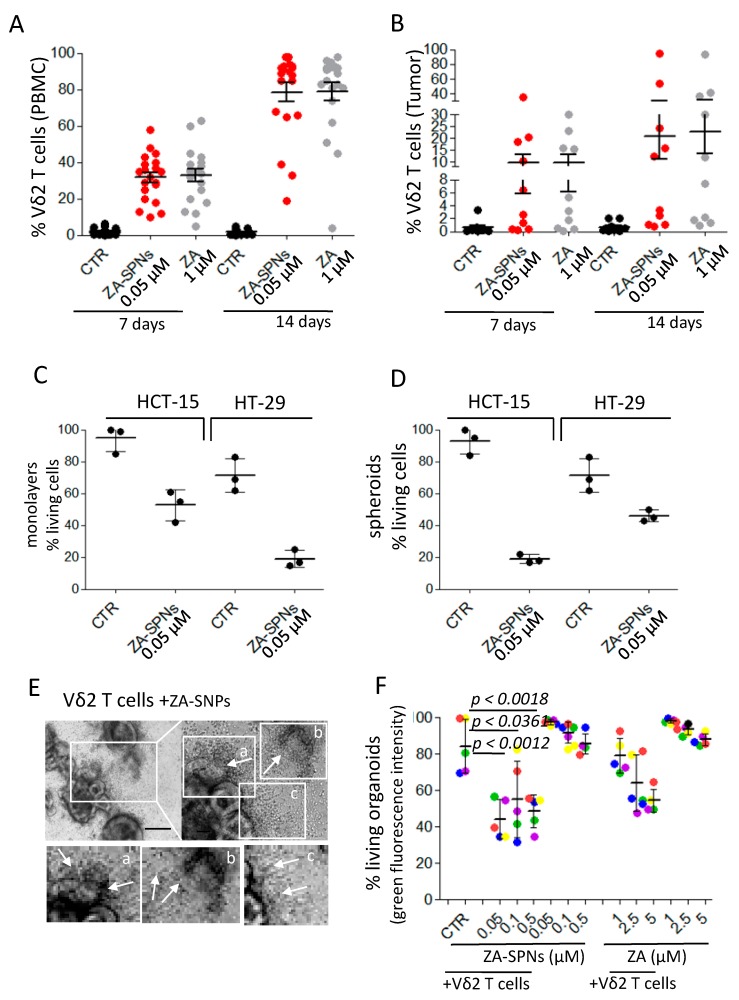
ZA-SPNs triggered the expansion of CRC patients’ Vδ2 T lymphocytes to be able to kill CRC spheroids and autologous organoids. (**A**) PBMC from CRC patients were cultured with ZA-SPNs (0.05 µM) or soluble ZA (1.0 µM) and 10 ng/mL (30 IU/mL) IL-2. The percentage of Vδ2 T lymphocytes was evaluated using FACS analysis on days 7 and 14. Vδ2 T cells in the starting PBMCs: 1.4 ± 1.6%, mean ± SD of 20 patients. (**B**) Tumor cell suspensions were cultured as in (A), and Vδ2 T lymphocyte expansion was determined on days 7 and 14. Results are expressed as percentages of Vδ2 T lymphocytes as mean ± SD of 10 patients’ specimens. Vδ2 T cells in the starting tumor-derived populations: 0.04 ± 0.05% (mean ± SD of 10 patients). (**C**,**D**) HCT-15 or HT-29 as adherent cells (C) or spheroids (D) were incubated with ZA-SPN-expanded Vδ2 T cells obtained from the PBMCs of three CRC patients. ZA-SPNs (0.05 µM) were further added or not (CTR) at the onset of the 48 h assay. E:T ratio was 3:1. Data expressed as a percentage of living CRC cells (crystal violet staining) as mean ± SD of six replicates/condition in three independent experiments. (**E**) Representative co-cultures of ZA-SPN-expanded Vδ2 effector T cells and the autologous OMCR-016TK organoids exposed to ZA-SPN. Upper-left image: 100×. Scale bar: 100 µm. Upper-right image: 200×. Lower panels: enlargements of quadrants a, b, and c in the upper-right panel. Asterisks in each panel: organoids. Arrows in each panel: Vδ2 T cells invading organoids (c, one almost destroyed in b) or in their neighborhood (c). (**F**) Organoids from five patients (OMCR18-006TK, green; OMCR18-006TK, red; OMCR19-006TK, blue; OMCR19-009TK, yellow; OMCR19-016TK, purple) were labeled with calcein-acteoxy methyl ester (AM) and challenged with autologous ZA-SPN-expanded Vδ2 T cells for 6 days without ZA-SPNs (CTR) or in the presence of ZA-SPNs or soluble ZA at the indicated concentrations. E:T ratio was 3:1. Fluorescence of each culture well was quantified using spectrofluorometry and compared to that of organoid cultures considered to be 100%. Data expressed as a percentage of green fluorescence as mean ± SD of three independent experiments with six replicates/condition. The *p*-values of ZA-SPNs versus CTR are shown.

**Figure 6 cancers-12-00104-f006:**
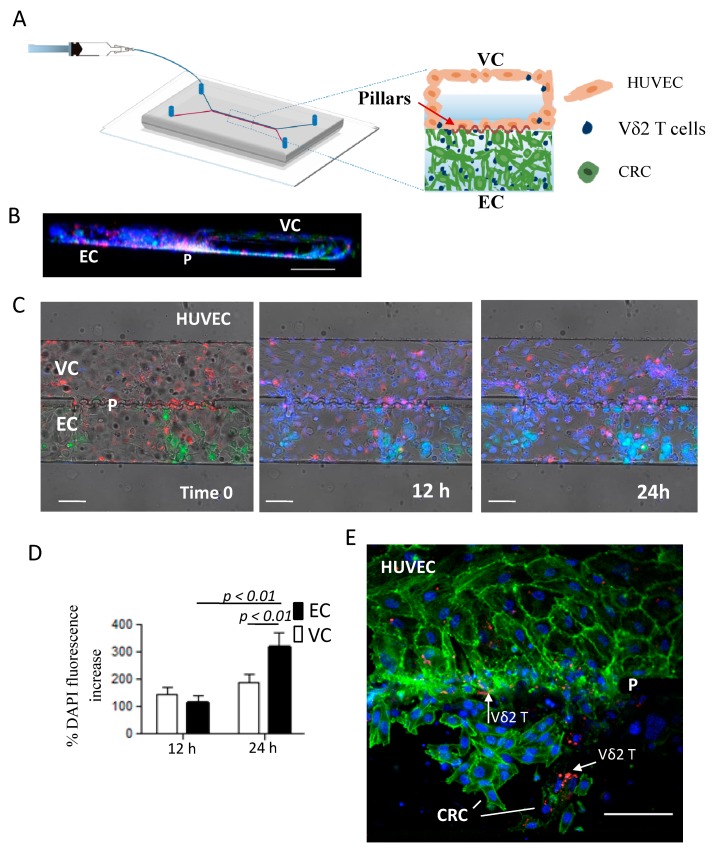
Transendothelial migration of ZA-SPN-expanded Vδ2 T lymphocytes toward CRC cells in a microfluidic system. (**A**) Double-chamber microfluidic chip [50] composed of a vascular compartment (VC), uniformly coated with human umbilical vascular endothelial cells (HUVECs), and an extravascular compartment (EC), hosting a Matrigel matrix with CRC cells. The two compartments were separated by an array of micropillars, equally spaced by 3 μm. ZA-SPN-expanded Vδ2 T cells were infused through the inlet into the VC, and their extravasation was mainly analyzed around the pillar area. (**B**) 3D reconstruction (lateral view) of the double channel microfluidic chip, showing on the right the HUVECs covering all of the channel (green/blue, phalloidin/DAPI) to mimic a complete vessel (VC); on the left, tumor cells (CRC, nuclear blue staining) in the EC. P: micropillar zone. Some Vδ2 T cells stained with CM-Dil (red) were visible in both compartments. Scale bar: 100 µm. (**C**) Time-lapse images following the localization of ZA-SPN-activated Vδ2 T lymphocytes before (time 0) and after (12 h, 24 h) injection into the VC. HUVEC membranes were stained using CM-Dil (red) and Vδ2 T cell nuclei were stained by DAPI (blue). EC: extravascular channel populated with GFP^+^ SW480 CRC cells (green). P: micropillars. Scale bar: 100 µm. (**D**) Quantification (Image J software) of DAPI fluorescence intensity increase measured at 12 h and 24 h compared to time 0 (addition of Vδ2 T cells) in the VC (white columns) or the EC (black columns). Data expressed as a percentage of fluorescence increase, compared to time 0; mean ± SE. The *p*-values are shown. (**E**) Representative confocal image (top view) showing Vδ2 T cells (red, CM-Dil, white arrows), close to HUVECs in the VC, and surrounding CRC cells in the EC. Phalloidin (green) stained the F-actin filaments to show the organization of HUVECs; actin filaments were also visible in the CRC. DAPI (blue) stained both the HUVEC and CRC nuclei. P: micropillar zone. Scale bar: 100 µm.

**Table 1 cancers-12-00104-t001:** In vitro effect (EC_50_) of ZA-SPNs and soluble ZA on Vδ2 T lymphocyte expansion.

Compound	Vδ2 Expansion ^a^	Vδ2 Fold Increase ^b^	Vδ2 Expansion from T + Mo ^c^
ZA-SPNs	1.355 × 10^−3^	2.067 × 10^−4^	7.096 × 10^−4^
Soluble ZA	3.685 × 10^−1^	4.2 × 10^−1^	5.449 × 10^−1^

^a^ EC_50_ value expressed as the µM concentration needed to reach 50% of the maximal percentage of Vδ2 T cells grown in culture starting from PBMCs. ^b^ EC_50_ value expressed as the µM concentration needed to reach 50% of maximal fold increase in Vδ2 T cell absolute number versus the initial Vδ2 T cell number. ^c^ EC_50_ value expressed as the µM concentration needed to reach 50% of the maximal percentage of Vδ2 T cells grown in culture starting from 10^5^ purified T lymphocytes with 10^3^ monocytes (Mo) added (T:Mo ratio 1:100). Results are the average of three replicates with a standard deviation of <10%.

**Table 2 cancers-12-00104-t002:** In vitro effect (EC_50_) of ZA-SPNs and soluble ZA on the Vδ2-mediated cytotoxicity of CRC cells and spheroids.

	SW620		HCT-15		HT-29	
Compound	Adh ^a^	Sph ^b^	Adh	Sph	Adh	Sph
ZA-SPNs	8 × 10^−2^	>6 × 10^−1^	1 × 10^−1^	5 × 10^−1^	1 × 10^−1^	5 × 10^−1^
ZA	1.2	1.0	1.0	1.1	1.7	1.0

^a^ The indicated CRC cell lines were cultured adherent to plates and used as targets (Adh). ^b^ CRC cell lines were used as spheroid targets (Sph) in ultra-low attachment plates. The EC_50_ value is expressed as the micromolar concentration needed to reach 50% of the maximal cytotoxic effect evaluated calculating the percentage of non-living cells. Results are the average of three determinations with a standard deviation of <10%.

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
