# Peer review of "Nanoformulated Zoledronic Acid Boosts the Vδ2 T Cell Immunotherapeutic Potential in Colorectal Cancer"

_cancers, 2019, doi:10.3390/cancers12010104_

Round 1

Reviewer 1 Report

Main concerns: 

- The experiment using autologous organoids and ZA-SPNs expanded T cells in Figure 5F is the most impactful of the manuscript and it is great to see the impact of ZA. However, the manuscript focuses on using nanoformulated-ZA over soluble ZA and it is therefore essential to conduct the same experiment with soluble ZA at its optimal concentration. It would be great if the infiltration of T cells within the organoids could be assessed as well as the engulfment of ZA-SPNs by either the organoids or the T cells. 

- The same is true for the extravasation experiment in Figure 6 that soluble ZA control should be used. Additionally, to confidently assess cell death of CRCs by the extravasated T cells, I recommend assessing with apoptotic markers such as active caspases. 

- It is argued in the manuscript that the ZA-SPN formulation performs better than soluble ZA in killing CRC cell lines in Figure 4. However, this becomes less convincing when using the spheroids of the CRC cell lines in Figure 4E. Additionally, in Supplementary Figure 2, there is no loss of viability when using the spheroids of the CRC cell lines when compared to the monolayer cultures, in the absence of T cells. This suggests that the direct effect of ZA-SPNs on CRC cells is limited however one should determine with the Cy5 labeled SPNs whether the spheroid cells engulf them. Also, this assay was performed for 48 hours only which according to previous data was just enough time to activate the T cells. One should extend the culture time to properly assess the tumour toxicity. 

- Where is the data of the other time points day 0, 7, 14 of T cell expansion for the experiment performed din Figure 2 A-B?

- Figure 3D-E: Are there replicates for the CRC cell line experiments for IPP production? Why was the starting number of cells halved? 

- Please include in supplementary the absolute number of Vd2 T cells in PBMC culture as well as in tumour suspensions after day 7 and day 14 in Figure 5A-B.

Minor comments concerning the text:

- Line 114: Text would benefit from adding the methodology of measuring the release of ZA from SPNs.

- Figure 1: there is no 'A-F' visible on the figure. 1A utilises acronyms in the figure that is not explained in the figure legend. 

- Include the concentrations of ZA-SPNs and soluble ZA that is tested within the body of the text. 

- Use the same scientific notation in Table I and II for the concentrations for ease of comparing. 

- Line 149: I believe it should refer to Figure 2A and not 1A.

- Figure 2A and 2E: the y-axis should be rewritten to reflect the data it is representing, e.g. % of Vd2 T cells in PBMC culture.

- Line 217: Please elaborate on why ZA-SPN is now used at 0.5 uM instead of the optimal 0.05uM as determined previously. 

- Please elaborate on why the accumulation of IPP in monocytes and CRC cell lines is beneficial for T cell growth, is the activation of monocytes necessary for the immune reaction against CRCs? It isn't completely clear in the argumentation why the experiments with monocytes is essential. 

- Figure 4: the methodology for these experiments are unclear. Are CRCs cultured for 24h with ZA-SPNs or for 4h followed by another 4h with T cells? 

- Line 322: Please clarify what is meant with tumour derived cell suspensions, are these intact tumours? 

- In the methods section it says the organoids are cultured at E:T ratio of 10:1 but in the figure legend of figure 5 it says 3:1.

Minor comments for discussion:

- Bone tropism of ZA is one of the stated reasons for pursuing the nano-formulation of ZA but is hardly discussed later on. 

- 100% of ZA is released from the nanoparticles by day 14, what is causing the expansion of Vd2 T cells after all the ZA is released? 

- If ZA was previously shown to stimulate the expansion of Vd2T cells as mentioned in the introduction, why does soluble ZA not lead to the proliferation of T cells in Figure 2C-E. 

- Why was the SW620 CRC cell line chosen to continue experiments with when LS180 CRC cell line was shown to produce more IPP and respond better to ZA-SPN? 

- In Figure 5F: Why do you think the % of living organoids isn't affected much by the 10-fold increase in ZA-SPNs when ZA-SPN expanded T cells are present, whereas the % of living organoids is reduced ~10% by the 10-fold increase in ZA-SPNs when ZA-SPN expanded T cells aren't present? 

Author Response

Review 1

The experiment using autologous organoids and ZA-SPNs expanded T cells in Figure 5F is the most impactful of the manuscript and it is great to see the impact of ZA. However, the manuscript focuses on using nanoformulated-ZA over soluble ZA and it is therefore essential to conduct the same experiment with soluble ZA at its optimal concentration. It would be great if the infiltration of T cells within the organoids could be assessed as well as the engulfment of ZA-SPNs by either the organoids or the T cells. 

We agree with the reviewer’s comments. First, data on the effects elicited by soluble ZA on CRC organoids have been added in the new version of Fig. 5F: it is now evident (and reported in the Results, page 13, lines 361-363) that ten to fifty-fold higher concentrations of soluble ZA are needed to obtain the same effects of ZA-SPNs. These data were already available, but we did not show them in the previous version of the manuscript to avoid possible misunderstandings in a quite complex figure. As suggested, we have analyzed the engulfment of ZA-SPNs by tumor spheroids, T lymphocytes and patients’ tumor organoids. We show that ZA-SPNs can enter both tumor tumor spheroids and organoids, besides T lymphocytes, and are visible inside the cells by 24h. These data have been added in an additional supplemental figure (Suppl. Fig.3 in the revised version) and described on page 10, line 275 and on page 13, lines 352-353. To get images with a good spatial resolution, infiltration of T cells into the organoids should be analyzed in time lapse over a long period of time with adequate instrumentation, such as a two-photon confocal microscope, to detect fluorescence in thick samples. We are planning these experiments in the near future. Nevertheless, we added a new figure (Suppl. Fig.7) in which the infiltration of tumor organoids with CFSE-labeled Vδ2 T cells is shown. Although images have been taken with a single photon confocal microscope, Vδ2 T cells infiltrating the crypt like organoids can be detected. Furthermore, the new version of Fig.5E (with enlargements added) of this paper shows that Vδ2 T lymphocytes can get in intimate contact with organoids and appear to invade one of them by destroying a portion of it.

- The same is true for the extravasation experiment in Figure 6 that soluble ZA control should be used. Additionally, to confidently assess cell death of CRCs by the extravasated T cells, I recommend assessing with apoptotic markers such as active caspases. 

The experiments suggested by the reviewer are of interest; indeed, until now, the relevance of ZA or ZA loaded nanoparticles as chemotatic agents has not been demonstrated. However, in the experiments depicted in Fig.6, Vδ2 T cells were used without the addition of ZA-SPNs during the assay. Indeed, in these experiments Vδ2 T cells expanded from PBMC with ZA-SPNs (named ZA-SPNs-expanded), on day 21 of culture were washed and put into the microfluidic system (this clarification has been added in the text, page 15 lines 392-393). Thus, no ZA-SPNs were present during the assay. The aim of these experiments was to demonstrate that Vδ2 T cells, originally stimulated with nanoparticles charged with ZA, can extravasate and interact with CRC in an artificial model, to reinforce the idea that these Vδ2 T cells can migrate through the endothelium and recognize tumor cells (page 15, lines 392-402).

Although altered actin organization can be considered a sign of initial cell damage, we agree with the reviewer’s concern on the need of an appropriate apoptotic marker to confirm that CRC cells are dying. However, to set up this experiment, at least three weeks time is needed because the limiting step is the adequate lining of the microfluidic system with endothelial cells. Due to the limited time allowed to reply to reviewers comments (10 days) we are not able to fully satisfy the reviewer. Thus, we deleted the words “as in dying cells” from lines 412-413 to avoid the overstatement that after the interaction with Vδ2 T cells the CRC cell line is dying.

- It is argued in the manuscript that the ZA-SPN formulation performs better than soluble ZA in killing CRC cell lines in Figure 4. However, this becomes less convincing when using the spheroids of the CRC cell lines in Figure 4E. Additionally, in Supplementary Figure 2, there is no loss of viability when using the spheroids of the CRC cell lines when compared to the monolayer cultures, in the absence of T cells. This suggests that the direct effect of ZA-SPNs on CRC cells is limited however one should determine with the Cy5 labeled SPNs whether the spheroid cells engulf them. Also, this assay was performed for 48 hours only which according to previous data was just enough time to activate the T cells. One should extend the culture time to properly assess the tumour toxicity. 

To address these points, we performed the suggested experiments with Cy5-ZA-SPNs and demonstrated that ZA-SPNs can enter the spheroids in about 24h (Suppl.Fig.3). This is now described in the Results section (page 10, line 275). Secondly, we assessed whether ZA-SPNs were cytotoxic to CRC tumor spheroids, at 72h and 96h of incubation. As neither soluble ZA nor ZA-SPNs were toxic to SW620, HCT-15 or HT-29 spheroids even after this period of incubation, we did not add any graph to Suppl.Fig.2, but this has been reported in the text as data not shown (Results, page 10, line 285). However, the main aim of this work was not to analyze the direct CRC cytotoxic effect of different ZA formulations but rather their capability of eliciting an immune response. Indeed, we determined whether ZA-SPNs were able to trigger Vδ2 T cell-mediated killing of tumor CRC cells; to this purpose, we used ZA-SPNs amounts that did not have an evident direct cytotoxic effect.

The cytotoxic effect of Vδ2 T cells on CRC spheroids exposed to ZA-SPNs was stronger than that elicited by ZA, with an EC50 of 0.5µM vs 1µM (Table II) and detectable already at 48h (Fig.4E) in the case of HCT-15 and HT-29 cell lines. SW620 appears to be more resistant, although able to engulf ZA-SPNs (Suppl.Fig.3A).

Possible explanations for the discordant effect of ZA-SPNs and soluble ZA on SW620 as adherent cells or spheroids, have been added to the Results section (page 10, lines 277-279) and in the Discussion (page 18, lines 478-484).

- Where is the data of the other time points day 0, 7, 14 of T cell expansion for the experiment performed din Figure 2 A-B?

To reply to this point, we inserted a new plot in Fig.2 (new panel A) where we show the kinetics of expansion of Vδ2 T cells from PBMC using 0.05µM ZA-SPNs and 1µM soluble ZA, the concentrations mostly efficient without direct toxic effects on T lymphocytes (this has been clarified on page 19, lines 569-571 and 578). Indeed, it is not easy to represent all the time points for all the different concentrations on the same plot (40 different points on one panel). Data are reported in the Results, page 6, lines 136-141.

- Figure 3D-E: Are there replicates for the CRC cell line experiments for IPP production? Why was the starting number of cells halved? 

IPP production by CRC cell lines depicted in Fig.3D and E refer to one representative experiment of two performed. This has been added to the legend. The number of CRC cells needed to perform the experiment is half of that required in the case of monocytes due to the ability of CRC cells to proliferate during the incubation time (CRC cells can almost reduplicate in 24h).

- Please include in supplementary the absolute number of Vd2 T cells in PBMC culture as well as in tumour suspensions after day 7 and day 14 in Figure 5A-B.

 These data have been included in the new Suppl.Fig.4.

Minor comments concerning the text:

- Line 114: Text would benefit from adding the methodology of measuring the release of ZA from SPNs.

The method to determine the free ZA from nanoparticles has been added (line 114, now 113).

- Figure 1: there is no 'A-F' visible on the figure. 1A utilises acronyms in the figure that is not explained in the figure legend. 

 Labels have been added to the figure and acronyms are now explained in the legend.

- Include the concentrations of ZA-SPNs and soluble ZA that is tested within the body of the text. 

We are not sure to understand this point. However, ZA-SPNs and ZA concentrations are indicated throughout the manuscript.

- Use the same scientific notation in Table I and II for the concentrations for ease of comparing. 

Scientific notation is now used also in Table II.

- Line 149: I believe it should refer to Figure 2A and not 1A.

 Actually, it is Fig.1A (Fig.1B of the revised version), just to recall that this concentration can be found also in Fig.1. The sentence has now been changed as follows: (5x10-2µM reported in Fig.1B).

- Figure 2A and 2E: the y-axis should be rewritten to reflect the data it is representing, e.g. % of Vd2 T cells in PBMC culture.

This correction has been done indicating on the y-axis: % Vδ2 T cells (PBMC).

- Line 217: Please elaborate on why ZA-SPN is now used at 0.5 uM instead of the optimal 0.05uM as determined previously. 

To detect IPP in cultured cells usually it is necessary an amount of ZA greater than that used to trigger optimal proliferation of Vδ2 cells. Indeed, it has been reported that tumor cell lines can produce detectable IPP using 10-50µM of ZA or other aminobisphosphonates (ref.10). These concentrations are adequate to detect IPP but they are also quite toxic. In our experiments, we used ten fold higher non-toxic amounts of ZA-SPNs than that used for Vδ2 cell stimulation to easily detect IPP with the method applied. It is clear from the same figure that there is a difference between the amounts of IPP detected in CRC cells using the 0.5µM and 0.05µM (panels D and E). Indeed, the IPP was barely detectable using the 0.05µM of ZA-SPNs. In the experiments performed with monocytes, we used ZA-SPNs and ZA at the same concentrations to verify whether nanoparticles were more potent than soluble ZA, as it can be supposed from the results shown in Fig.2 and 3A and B.

- Please elaborate on why the accumulation of IPP in monocytes and CRC cell lines is beneficial for T cell growth, is the activation of monocytes necessary for the immune reaction against CRCs? It isn't completely clear in the argumentation why the experiments with monocytes is essential. 

It is well established that monocytes (Mo )are the main cell population involved in N-BPs-mediated triggering of Vδ2T cells [6,20,21]. This was already stated on page 8, lines 208-209, but we added a sentence to clarify the concept that this effect is due to IPP production (lines 209-210).

- Figure 4: the methodology for these experiments are unclear. Are CRCs cultured for 24h with ZA-SPNs or for 4h followed by another 4h with T cells? 

Panels A-C: CRC incubated for 4h with ZA-SPNs then T cells were added to CRCs engulfed with ZA-SPNs for additional 4h. This has been clarified on page 9, lines 250-252.

- Line 322: Please clarify what is meant with tumour derived cell suspensions, are these intact tumours? 

 Cells were isolated from tumor specimens by enzymatic digestion as described in Material and Methods and used in these experiments. The sentence has been changed into: …mononuclear cells isolated from the tumor..(page 12, line 326)

- In the methods section it says the organoids are cultured at E:T ratio of 10:1 but in the figure legend of figure 5 it says 3:1.

We apologize, 10:1 was used in preliminary experiments, the right ratio is 3:1 and has been corrected in Material and Methods (page 21, line 675).

Minor comments for discussion:

- Bone tropism of ZA is one of the stated reasons for pursuing the nano-formulation of ZA but is hardly discussed later on. 

Encapsulating ZA into SPNs should provide also the advantage of a reduction of bone tropism due to the EPR effect: this was already explained in the Introduction (page 3, lines 76-81) and discussed on page 17, lines 442-444. Two sentences on this issue have been added to the discussion (page 17, lines 444-445 and 450-452). Other evidences come from the literature in the murine model (ref.53, already cited) and we added comments on this point (page 17, lines 469-471).

- 100% of ZA is released from the nanoparticles by day 14, what is causing the expansion of Vd2 T cells after all the ZA is released? 

The expansion is due to the progressive metabolic effect of ZA on mevalonate pathway. Once activated, cell proliferation goes on, provided IL2 is present (by the first 48-72h) or added. This point has been discussed on page 17, lines 450-452.

- If ZA was previously shown to stimulate the expansion of Vd2T cells as mentioned in the introduction, why does soluble ZA not lead to the proliferation of T cells in Figure 2C-E. 

At variance with ZA-SPNs, ZA does not work in the absence of IL2, while it can be efficient in inducing expansion of Vδ2 T cells from PBMC in the presence of IL2, as shown in Fig.2A-B and 5A. This point was already discussed on page 17, lines 453-460.

- Why was the SW620 CRC cell line chosen to continue experiments with when LS180 CRC cell line was shown to produce more IPP and respond better to ZA-SPN? 

This cell line SW620 was chosen because LS180 cell line does not form spheroids, as we previously reported (ref.45). Also, the aim was to verify whether a rather insensitive cell line, producing lower amounts of IPP (page 8, lines 226-227), could respond better when stimulated with ZA-SPNs.

- In Figure 5F: Why do you think the % of living organoids isn't affected much by the 10-fold increase in ZA-SPNs when ZA-SPN expanded T cells are present, whereas the % of living organoids is reduced ~10% by the 10-fold increase in ZA-SPNs when ZA-SPN expanded T cells aren't present? 

In the presence of ZA-SPNs-expanded Vδ2 T lymphocytes, organoid vitality decreased by 10-30% due to the cytotoxic effect of these lymphocytes. When ZA-SPNs were added to the cytolytic assay, they increased the anti-tumor efficiency of ZA-SPNs-expanded Vδ2 T lymphocytes, leading to organoid vitality reduction of 50%. The addition of soluble ZA did not produce the same effect, unless 10 to fifty-fold higher concentrations were used. This was clarified on page13, lines 358-362.

Reviewer 2 Report

The study by Mascolo et al evaluated the efficacy of ZA-SPN in boosting Vδ2T cell responses in CRC. The authors show that ZA-SPNs induce proliferation and activation of Vδ2 T cells up to three orders of magnitude more efficiently than soluble ZA. Moreover, they show that these activated Vδ2 T cells were able to kill CRC cells and patient derived spheroids. Though this study has merit, there are some recommendations/concerns which the authors should address.

The authors pointed out that one of the main limitations in using free ZA for the treatment of CRC was is its bone tropism, however authors didn’t prove this with their ZA-SPN in vivo mouse models. Though the authors used organoid model and pointed out as limitation in discussion section, it would be imperative in future to test efficacy of ZA-SPNs in humanized immune models of sporadic CRC.    

In Figure 2A and 2B, the author’s report that 5uM ZA dose was needed to obtain the same effect of ZA-SPNs of 0.05uM. However, in figure 5A and 5B, the authors report that only 1uM of ZA was needed to obtain similar effect as ZA-SPNs of 0.05uM. Can the authors explain this difference?

Moreover, when the authors used 1uM of ZA in Figure 2D-2E, they found no significant proliferation of Vδ2 T lymphocytes with soluble ZA while using the same concentration of 1uM of ZA in Figure 5A-5B, the authors report to induce proliferation of Vδ2 T lymphocytes. Why these discrepancies were observed.

Moreover, the authors report that ZA-SPNs were able to expand Vδ2 T lymphocytes in the absence of exogenous growth factors via neo expression of activation antigens in Fig 2F. The authors should also test this in Figure 5A without the use of IL-2 and show the same impact as observed in Fig 2F.

The authors report to incubate purified T lymphocytes with monocytes for 21 days. As half – life of monocyte is short; would like to know the feasibility of these experiments.

Author Response

Review 2

The study by Mascolo et al evaluated the efficacy of ZA-SPN in boosting Vδ2T cell responses in CRC. The authors show that ZA-SPNs induce proliferation and activation of Vδ2 T cells up to three orders of magnitude more efficiently than soluble ZA. Moreover, they show that these activated Vδ2 T cells were able to kill CRC cells and patient derived spheroids. Though this study has merit, there are some recommendations/concerns which the authors should address.

The authors pointed out that one of the main limitations in using free ZA for the treatment of CRC was is its bone tropism, however authors didn’t prove this with their ZA-SPN in vivo mouse models. Though the authors used organoid model and pointed out as limitation in discussion section, it would be imperative in future to test efficacy of ZA-SPNs in humanized immune models of sporadic CRC.    

We agree with the referee that it is important to check the tumor tropism of these ZA-SPNs in the animal model, before considering this formulation as potential therapeutic tool, and we take her/his advise for future experimental plans. Nevertheless, the preferential tumor localization of SPNs due to the EPR effect, that can prevent ZA leakage before time and its natural accumulation in bone tissue, has been reported by the authors and others (refs. 33, 34,38) and recently in the mouse model as well (ref.51). This is now discussed in more detail on page 17, lines 444-445 and 467-471.

In Figure 2A and 2B, the author’s report that 5uM ZA dose was needed to obtain the same effect of ZA-SPNs of 0.05uM. However, in figure 5A and 5B, the authors report that only 1uM of ZA was needed to obtain similar effect as ZA-SPNs of 0.05uM. Can the authors explain this difference?

Moreover, when the authors used 1uM of ZA in Figure 2D-2E, they found no significant proliferation of Vδ2 T lymphocytes with soluble ZA while using the same concentration of 1uM of ZA in Figure 5A-5B, the authors report to induce proliferation of Vδ2 T lymphocytes. Why these discrepancies were observed.

We used 1 µM ZA concentration to stimulate patients’ PBMC and specimens, since using ZA at 5 µM concentration on heathy donors PBMC, about 20% of dead cells was detected among Vδ2 T cells after 14d of culture, while the percentage of dead cells with 1 µM ZA was negligible. In any case, data are always referred to percentages of cells analized by FACS calculated gating viable cells (page 20, line 592). This is now clarified in Materials and Methods, page 19, lines 571-572.

From the new panel A of figure 2, it is evident that the 5µM or 1 µM concentrations of soluble ZA can trigger very similar expansion of Vδ2 T cells when exogeneous IL2 is added to the cell cultures (this is the classical method to get the expansion of Vδ2 T cells from PBMC as reported in the paper and literature).

From the figure 2B, left panel (panel A of the old version of the manuscript), it was already evident that 5µM or 1 µM concentrations of soluble ZA can induce similar expansion of Vδ2 T cells from PBMC (please take into account of the Log scale of the x-axis to correctly identify these two concentrations). Again, these experiments have been performed with the addition of IL2 to cell cultures. These culture conditions are the same applied for patients as depicted in panels 5A and 5B. Thus, there is no discrepancies between what reported in Fig.2 and Fig.5. it is of note that, panels 2D and 2E, are referred to the proliferation of Vδ2 T cells without adding exogeneous IL2. In this peculiar experimental situation, soluble ZA is not working while ZA-SPNs are effective. This to remark the stronger effect exerted by the nanoformulated ZA.

In addition, experiments reported in Fig.2A and B, were performed with PBMC from healthy donors, while data in Fig. 5A come from experiments with patient’s PBMC. A certain degree of heterogeneity in the response to ZA is reported and found also in the present work (ref. 21 and see Fig. 2C), in particular when lymphocytes isolated from the tumor are used (Fig.5B).

Moreover, the authors report that ZA-SPNs were able to expand Vδ2 T lymphocytes in the absence of exogenous growth factors via neo expression of activation antigens in Fig 2F. The authors should also test this in Figure 5A without the use of IL-2 and show the same impact as observed in Fig 2F.

This experiment has been already done; however, in the case of patients’ PBMC it was not always possible to get reproducible results. This might be due to the fact that patients’ PBMC are freezed and thawed or that they are in a different activation state that does not allow the sequential expression of CD69 and CD25. Also, as mentioned above, heterogeneity in patients’ response to ZA was found. Moreover, in the case of healthy donors’ PBMC, only a small fraction (~17%) of Vδ2 T lymphocytes could actually express intracytoplasmic IL-2 upon stimulation with ZA-SPNs thus being responsible for a sort of paracrine loop that sensitizes Vδ2 T cells to cytokines and growth factors present in the extracellular milieu (page 17, lines 457-460).

The authors report to incubate purified T lymphocytes with monocytes for 21 days. As half – life of monocyte is short; would like to know the feasibility of these experiments.

We agree with the reviewer: however, monocytes are needed in the first few days to produce IPP, and the number of γδT lymphocytes was evaluated at day 21. The sentence on page 8, lane 211 has been corrected.

Reviewer 3 Report

Dear author,

I have read the article, 'Nanoformulated-Zoledronic Acid Boosts Vδ2T Cell
Immunotherapeutic Potential in Colorectal Cancer' with high interest. THe article is novel and the experiments are well carried out. Hence I am recommending to accept this manuscript as its present form.

Minor comment: please do English, grammatical and formatting correction throughout the manuscript as there are many small mistakes.

Author Response

I have read the article, 'Nanoformulated-Zoledronic Acid Boosts Vδ2T Cell
Immunotherapeutic Potential in Colorectal Cancer' with high interest. THe article is novel and the experiments are well carried out. Hence I am recommending to accept this manuscript as its present form.

Minor comment: please do English, grammatical and formatting correction throughout the manuscript as there are many small mistakes.

We are pleased for the positive judgement of the reviewer. As requested, the manuscript has been carefully revised for misprints and mistakes.

Round 2

Reviewer 2 Report

The authors have addressed the requested concerns and I don’t have any questions further.